Report

EMBO
reports

# Embryonic stem cells maintain high origin activity and slow forks to coordinate replication with cell cycle progression

Kiminori Kurashima [ID] [1,2], Yasunao Kamikawa [ID] [1,3] & Tomomi Tsubouchi [ID] [1,2 ✉]

## Abstract

Embryonic stem (ES) cells are pluripotent stem cells that can produce all cell types of an organism. ES cells proliferate rapidly and are thought to experience high levels of intrinsic replication stress. Here, by investigating replication fork dynamics in sub-stages of S phase, we show that mammalian pluripotent stem cells maintain a slow fork speed and high active origin density throughout the S phase, with little sign of fork pausing. In contrast, the fork speed of non-pluripotent cells is slow at the beginning of S phase, accompanied by increased fork pausing, but thereafter fork pausing rates decline and fork speed rates accelerate in an ATR-dependent manner. Thus, replication fork dynamics within the S phase are distinct between ES and non-ES cells. Nucleoside addition can accelerate fork speed and reduce origin density. However, this causes miscoordination between the completion of DNA replication and cell cycle progression, leading to genome instability. Our study indicates that fork slowing in the pluripotent stem cells is an integral aspect of DNA replication.

**Keywords** Pluripotent Stem Cells; Replication Fork Speed; Origin Density; Cell Cycle Regulation
**Subject Categories** DNA Replication, Recombination & Repair; Stem Cells & Regenerative Medicine

## Introduction

Embryonic stem (ES) cells are pluripotent stem cells derived from early-stage embryos and are highly proliferative with a shortened G1 phase. Molecularly, these cell cycle profiles are characterized by the expression of high levels of cyclins, Cdk1 and Cdk2 kinases, along with the reduced expression of their inhibitors (Fujii-Yamamoto et al, 2005; Liu et al, 2019; Stead et al, 2002; Tsubouchi and Fisher, 2013; Wang et al, 2017). The maintenance of a short G1 is important for pluripotency (Calder et al, 2013; Coronado et al,

2013; Pauklin and Vallier, 2013), but a short G1 is also thought to cause replication stress, resulting in the reduction of replication fork speed (Ahuja et al, 2016). Nevertheless, ES cells can be indefinitely cultured without presenting significant genome instability (Tichy and Stambrook, 2008). Ultra-fine bridges (UFBs) are prominent features in cells exposed to sources of replication stress and experiencing delayed replication (Chan and Hickson, 2011; Chan et al, 2007). Consistently, ES cells present relatively low level of UFBs during mitosis compared MEFs (Ahuja et al, 2016). A recent report investigating totipotent cells in the early embryo and totipotent-like 2-cell-like cells (2CLCs) that arise spontaneously from ES cells indicated that these cells present even slower fork progression than ES cells, and showed that slow replication forks regulate cellular plasticity (Nakatani et al, 2022). Thus, slow replication forks in ES cells may also be an integral aspect of pluripotency and may represent a unique DNA replication mechanism that tolerates genome instability.

Genome-wide DNA replication is highly regulated to ensure that the entire genome is duplicated in a timely and accurate fashion before committing to mitosis. The amount of DNA synthesized within a set amount of time per genome is determined by the rate of DNA synthesis per replication fork (i.e., fork speed) and the number of replicating forks present within the genome. Several studies suggest cross-talk between the speed of replication forks and the activation of origins. For instance, when replication forks slow down or arrest in the presence of polymerase inhibitors or due to a shortage of dNTPs, they are backed up by additional origin firing (Alver et al, 2014; Técher et al, 2017; Woodward et al, 2006). On the other hand, inefficient origin firing caused by the suppression of components of the pre-initiation complex leads to an increase in replication fork speed (Moiseeva et al, 2017). However, whether the optimal replication fork speed and the number of active origins is conserved across cell types, or is set up differently in different cell types, is unknown. In the former scenario, for example, fork speeds deviating from the usual ranges indicate that they suffer from unwanted replication stress. If latter is the case, different cell types operate replication forks to progress at a certain rate, to suit their cell-type-specific needs and functions. In other words, slow forks do not necessarily link to the level of replication stress.

[1]Laboratory of Stem Cell Biology, National Institute for Basic Biology, National Institutes of Natural Sciences, Okazaki, Japan. [2]Department of Basic Biology, SOKENDAI (The Graduate University for Advanced Studies), Okazaki, Japan. [3]Present address: Department of Biochemistry, Institute of Biomedical & Health Sciences, Hiroshima University, Hiroshima, Japan. ✉E-mail: ttsubo@nibb.ac.jp

Within a single S phase, DNA replication progresses through genomic regions with different properties. DNA replication during early S phase typically starts in transcriptionally-active open chromatin regions, and then moves to the nuclear/nucleolar periphery, before reaching the heterochromatic silent regions (Chagin et al, 2016; Nakamura et al, 1986; O'Keefe et al, 1992). Heterochromatic regions are presumed "hard to replicate", because of their closed chromatin structure. However, the extent to which replication forks encounter impediments during late S phase in unperturbed culturing conditions remains incompletely explored.

In this study, we sought to understand the significance of replication fork dynamics unique to ES cells especially in the context of genome stability. By molecularly investigating replication fork dynamics in substages of S phase, we found that pluripotent stem cells present consistently slow replication forks throughout S phase with little sign of fork pausing or arrest. However, forced acceleration of replication forks caused a reduction in the number of active origins throughout S phase and fork pausing/arrest specifically in late S phase, which led to an increase in the frequency of UFB and micronuclei formation. Our results show that reduced replication fork speeds in ES cells is an integral aspect of replication dynamics employed by fast-cycling pluripotent cells. We propose that a high number of active origins and slowly progressing replication forks are key for timely genome replication in pluripotent stem cells.

## Results and discussion

### DNA replication fork dynamics within the S phase distinguish pluripotent and non-pluripotent cells

DNA replication forks are reported to progress slowly in mouse embryonic stem (ES) cells compared to their non-pluripotent counterparts (Ahuja et al, 2016). Using the DNA fiber assay, we found that the replication fork speed in the E14Tg2A mouse ES cell line was indeed slower compared to the SNL mouse embryonic fibroblast (MEF) line (Fig. 1A). In addition, we noted that the replication fork speed in MEFs was more variable compared to ES cells. The slow fork speed observed in ES cells is not caused by differences in the medium components or active transcription (Fig. EV1A–C; Appendix Fig. S1A). Reactive oxygen species (ROS) are also reported to cause fork slowing (Huang et al, 2016; Somyajit et al, 2017), but the addition of ROS scavenger *N*-acetyl-l-cysteine (NAC) did not increase the replication fork speed in ES cells to the level of MEFs (Fig. EV1D).

To investigate whether fork speed varies within the S phase, we sorted cells based on DNA content and measured replication fork speed in each population (Fig. 1B,C). Fixation methods prior to cell sorting did not affect fork speed measurements (Fig. EV1E). Here, in MEFs, replication fork speed is slow in the early S phase, but increases in late S-phase populations (Fig. 1D). On the other hand, replication fork speed remained low throughout the S phase in ES cells (Fig. 1D). MEF-like fork speed dynamics were also observed in other mouse and human non-pluripotent cell lines such as neuron cell-derived neuro2A (N2A), mouse B lymphoblastoid cells (mB), HeLa and RPE (Figs. 1E,F and EV1F). Human induced pluripotent cells (hiPS), showed less-variable and slow fork speed across substages of S phase as in mouse ES cells (Fig. 1E,F). Thus, slow

forks appeared to be a common feature for both mouse and human pluripotent stem cells. To confirm that these fork dynamics are indeed linked to pluripotency, we induced mouse ES cells to differentiate by removing LIF from the media (Fig. EV1G). Replication fork speed increased in late S but not in early S, presenting a profile similar to non-pluripotent cells (Fig. 1E,F, right panels).

### MEFs experience frequent fork pausing in the early S phase, whereas ES cells experience less fork pausing throughout S phase

Various sources of replication stress can lead to fork pausing or collapse, which appear as overall fork slowdown. To assess replication fork pausing frequency while minimizing variations caused by differences in local features such as chromosome structure and DNA sequences, we investigated the relative speed of the two replication forks emanating from a single newly fired origin (i.e., scored "fork symmetry") (Fig. 2A). We defined forks as "symmetric" when the difference in lengths of a pair of labeled forks is 30% or less. In MEFs, we noticed that the early S-phase fractions showed slightly higher (26.8%) fork "asymmetry" compared to late S (18.0%) (Figs. 2B and EV2A; MEF). In contrast, ES cells showed a much lower percentage of asymmetric forks in both early and late S phases (13.0% and 12.0%, respectively) (Figs. 2B and EV2A; ES). Longer pulse-labeling did not significantly increase the fork asymmetry rate in ES cells (Fig. EV2B). Thus, the measured lower fork asymmetry rate in ES cells is not a result of underestimation due to insufficient pulse-labeling time. Indeed, when ES cells were treated with hydroxyurea (HU), which causes dNTP shortage and imbalance, we detected an increase in replication fork asymmetry (10.0% vs 19.2%) (Figs. 2C and EV2C). Interestingly, HU increased replication fork asymmetry to different levels in early vs late S phases in MEFs (Fig. 2D). We also noted that the fork asymmetry rate does not correlate directly with the fork speed, indicating that the fork speed alone does not predict the level of fork pausing frequencies. Fork asymmetry is observed even in a condition (i.e., 10 μM HU) where overall fork slowdown is insignificant, and thus serves as a sensitive assay to detect fork pausing events. Taken together, our results indicate that ES cells do not experience a high level of replication stress that causes significant fork pausing or arrest.

Fork slowing can occur via activation of high number of origins (Moiseeva et al, 2017). To test if the primary cause of fork slowing in ES cells is related to origin activity, we used an assay reported previously (Rodriguez-Acebes et al, 2018). By using a CDC7 inhibitor (CDC7i) that reduces origin activity, we can ask whether the primary cause of fork slowing is simply the high number of active origins or is linked to the fork itself. Only if the primary cause of fork slowing is the high number of active origins, then CDC7i should allow acceleration of the fork speed. We confirmed that CDC7i reduced the origin density, as evidenced by increased inter-origin distances (IODs) (Fig. 2E; Appendix Fig. S1B). Short IODs reflect higher origin density and long IODs reflect reduced firing rate. At the same time, CDC7i significantly increased replication fork speed in ES cells. These results strongly suggest that the fork dynamics that differentiate pluripotent and non-pluripotent cells originate from their different origin activities, and not from the level of replication stress.

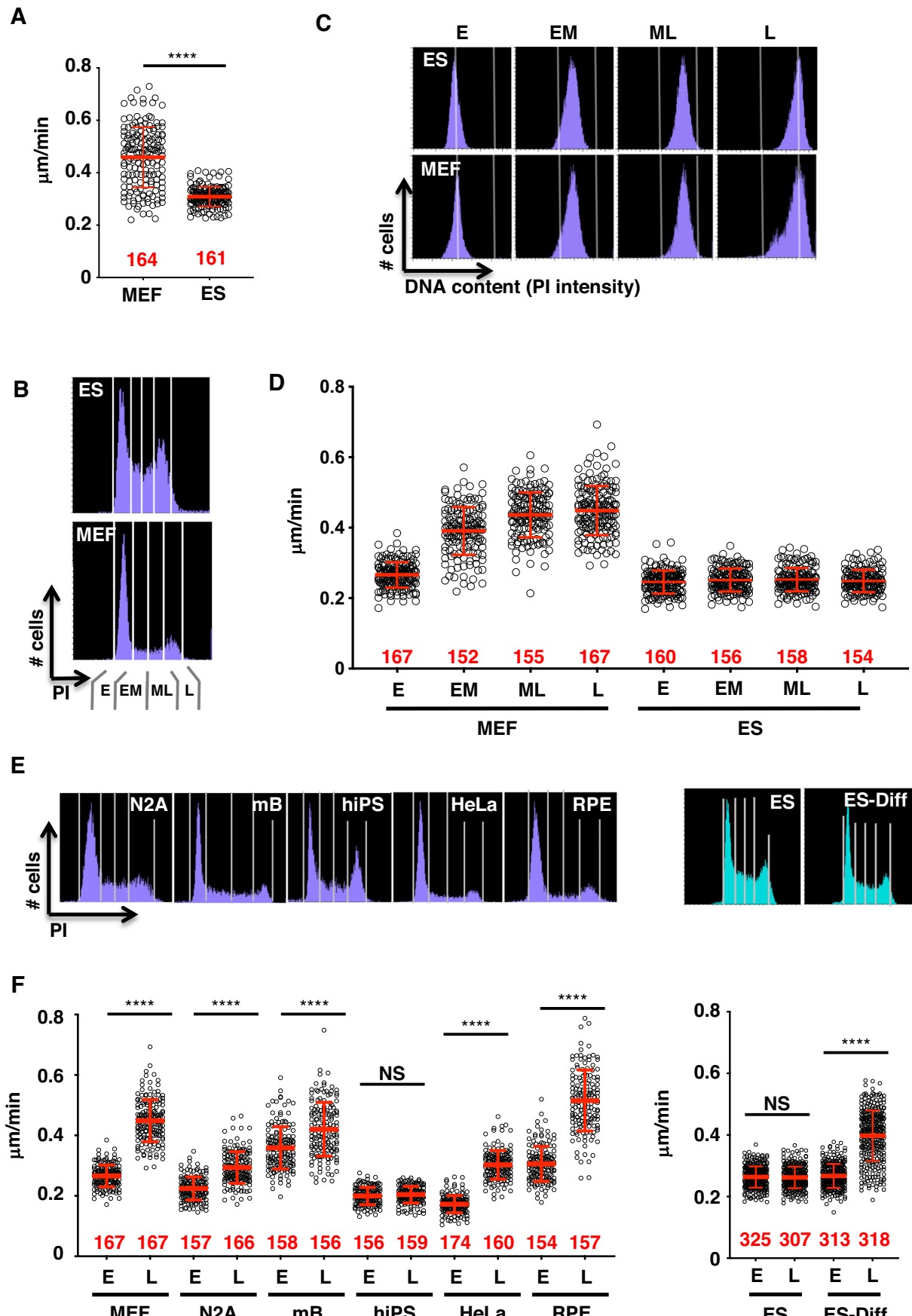

◄  **Figure 1.  Pluripotent and non-pluripotent cells exhibit different replication fork speed dynamics during the S phase.**

(A) Asynchronously growing mouse embryonic fibroblasts (MEF) and mouse embryonic stem cells (ES) were sequentially labeled with IdU and CldU, resuspended in PBS, and were subjected to DNA fiber assay to measure fork speed (methods). (B–D) Replication fork speed in different substages of S phase in MEF and ES cells. Cells were sorted based on DNA contents [using propidium iodide (PI) staining] using the gates indicated (E: Early, EM: Early Mid, ML: Mid Late, L: Late) in (B). (C) PI profiles of sorted fractions were reanalyzed. (D) Fork speed in fractions enriched for different substages of the S phase was measured as in (A). (E) PI profiles and gates used for cell sorting. (F) Various mouse and human cell lines were sorted as in (E), and the fork speeds were measured as in (A). In the right panels, ES cells were differentiated by removing LIF from the media for 3 days (ES-Diff), collected and sorted for fiber assay along with ES cells in the normal growing condition (ES). The numbers of scored forks are indicated in red. Data are presented as mean +/− SD. NS, not significant ($P \geqq 0.05$), ****$P < 0.0001$, Mann–Whitney tests (A, F). Data from a representative experiment is shown ($n = 2$ independent experiments). Source data, including all measurements and exact $P$ values are available. Source data are available online for this figure.

## ATR/Chk1 is responsible for fork acceleration in MEFs

ATR plays a central role in signaling DNA replication stress caused by both endogenous and exogenous sources to accurately complete genome replication (Hustedt et al, 2013; Saldivar et al, 2017). Because MEFs present higher levels of fork asymmetry in early S phase (Fig. 2D), we wondered if ATR is activated in response to paused forks and modulate fork dynamics in MEFs.

Specifically, if ATR activation is responsible for the acceleration of replication fork speed in MEFs, ATR inhibition should abolish the effect. Indeed, an ATR inhibitor (ATRi) abolished fork acceleration in MEFs (Fig. 3A, left). The overall fork speed when cells were treated with the ATRi was reduced, likely as a result of an increase in the number of fired replication origins, as reported previously (Moiseeva et al, 2017; Moiseeva et al, 2019; Petermann et al, 2010; Syljuåsen et al, 2005). In ES cells, ATRi reduced overall fork speed, but did not affect relative fork speeds in early vs late S phase (Fig. 3A, right). Thus, ATR does not play a role in accelerating fork speed within the S phase in ES cells, although being active and present. MEFs may enter S phase before reaching optimal condition for DNA replication, triggering the activation of the MEC1/ATR pathway to induce fork acceleration, similar to findings in yeast (Forey et al, 2020). Although the exact role of ATR in fork acceleration in MEFs remains to be uncovered, the same system clearly does not operate in ES cells.

## Exogenous ribonucleoside supplementation accelerates replication fork speed throughout the S phase in ES cells and in the early S phase of MEFs

To gain insight into the biological significance of reduced fork speed in ES cells, we sought to force an increase of replication fork speed and investigate its impact. For this, we used ribonucleosides, which are commonly used reagents to induce replication fork acceleration (Aird et al, 2013; Bester et al, 2011; Garzón et al, 2017; Halliwell et al, 2020; Kotsantis et al, 2016).

In both MEFs and ES cells, treatment with nucleosides increased overall fork speed, although the difference was more subtle in MEFs (Fig. 3B). We further investigated the effects in different substages of S phase. Interestingly, the replication fork sped up only in the early S phase and not in late S in MEFs (Fig. 3C). On the other hand, an increase was evident in both early- and late- S fractions of ES cells (Fig. 3C).

The increase in the fork speed in ES cells is not a result of their differentiation, as continuous nucleoside supplementation for as long as 12 days neither altered colony morphology

(Fig. EV3A) nor the expression of pluripotency-associated genes (Fig. EV3B).

## Exogenous ribonucleoside supplementation reduces ATR activity in MEFs

To evaluate the level of ATR activity, we measured the abundance of phosphorylated Rad17 (pRad17) as a measure of ATR kinase activity. The addition of the ATR inhibitor significantly reduced, and the addition of HU increased, pRad17 in both MEFs and ES cells as assayed by immunofluorescence (IF) and western blotting, confirming that pRad17 serves as a marker for ATR activity (Figs. 3D and EV3C,D; Appendix Fig. S1C). Importantly, the addition of nucleosides reduced pRad17 levels in MEFs (Figs. 3D and EV3C,D; Appendix Fig. S1C), consistent with the idea that nucleoside addition reduces the level of replication stress. Indeed, fork asymmetry in the early S phase in the presence of nucleosides was substantially reduced (15.1%, vs 26.8%) (Figs. 2B and 3E). In ES cells, nucleoside addition slightly increased pRad17 levels (Figs. 3D and EV3C,D), but did not detect a significant change in fork asymmetry (13.0% vs 13.5% in early S) (Figs. 2B and 3E).

## Replication fork acceleration causes fork pausing specifically in the late S phase

With nucleoside supplementation, fork asymmetry increased in the late S phase in both MEF and ES cells. Nucleosides affected ES cells more drastically in ES cells, both in terms of the percentage of forks outside the 30% range (33.9% in MEF vs 53.7% in ES) and the difference in the length of the two labeled forks (i.e., more dots are further away from the symmetric range) (Fig. 3E). The overall replication fork speeds in early vs late S phase are roughly the same with nucleoside addition (Fig. 3C), suggesting that it is not the fork speed per se that leads to fork pausing/arrest. γH2AX increased from 6.1 to 13.1%, suggesting the presence of DNA damage and/or replication stress (Fig. EV3E). Transcription activity is not the cause of fork asymmetry even in the presence of nucleosides, as the fork asymmetry rate did not change in the presence of transcription inhibitors (Fig. EV3F).

A recent report using human pluripotent stem cells showed that nucleosides reduce replication stress and enhance their survival (Halliwell et al, 2020). We evaluated the effect of varying concentrations of nucleoside supplementation on fork asymmetry in a human iPS cell line (PChiPS771, REPROCELL), but did not observe reduction in fork asymmetry in this cell line in the media condition we used (Fig. EV3G). Thus, in our hands, nucleoside

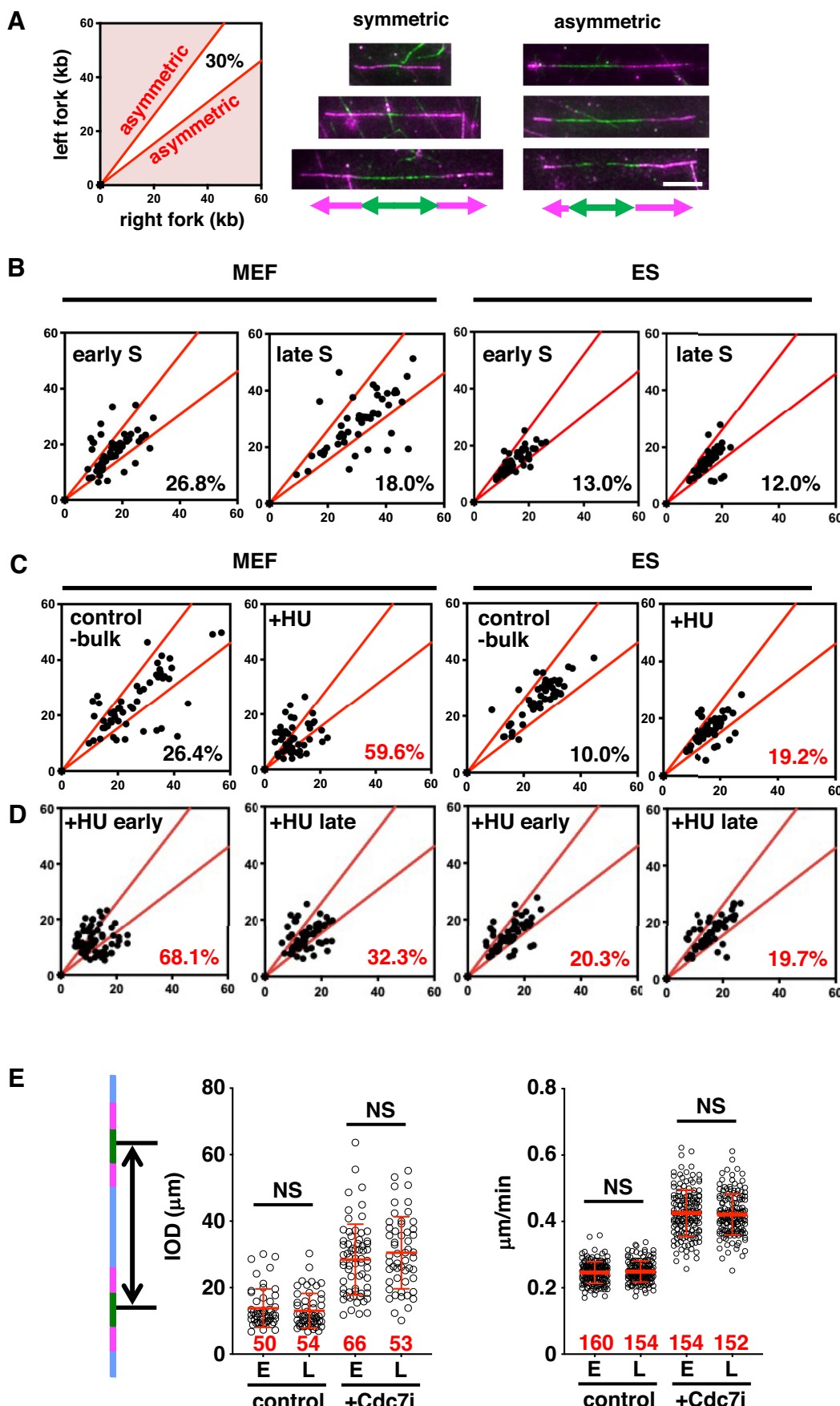

◀ **Figure 2.** Mouse ES cells exhibit minimal fork pausing both in the early- and late S phase, while MEFs exhibit higher basal levels of fork pausing in early S phase.

(A) Explanation of the fork asymmetry plot and representative images of stained DNA fibers used for the fork asymmetry analyses. Those with a larger than 30% difference in length were scored as "asymmetric". The percentage of asymmetric forks among the total forks scored is indicated at the bottom right corner of each plot in (B, C). Bar, 10 µm. (B) Fork asymmetry rates scored for early and late S phase fractions in MEF and ES cells. (C) Fork asymmetry rates in the presence or absence of HU. For this experiment, cells were first labeled with IdU for 15 min, then CldU and HU (100 µM final) were added at the same time for 30 min. (D) Cells labeled as in (C) were fixed and sorted to score fork asymmetry in early and late S fractions. $n = 3$ independent experiments for (B–D), and a representative data is shown. Exact numbers of scored fibers and measurements are available in the source data. (E) ES cells were treated with or without Cdc7i (10 µM final for 2 h) and were sorted for early (E) and late (L) fractions to measure inter-origin distance (IOD; left) and fork speed (right). The numbers of scored forks are indicated in red. Data from a representative experiment is shown ($n = 2$ independent experiments). NS not significant ($P \geqq 0.05$), Mann–Whitney tests. Source data including all measurements and exact $P$ values are available. Source data are available online for this figure.

supplementation presents a consistent outcome in both human and mouse pluripotent lines.

## Replication fork acceleration leads to reduced origin activation

Nucleoside supplementation significantly accelerated replication fork speed in ES cells, but the overall growth rate was not affected. We investigated the effect of nucleoside addition on the overall S phase duration by live imaging, using endogenous PCNA tagged with mNeonGreen as a marker for ongoing DNA replication. S-phase durations, as determined by the time when cells showed PCNA foci, remained similar with or without nucleosides (Figs. 4A, left and EV4A,B). Curiously, the duration of G2 phase shortened specifically with nucleoside addition (Figs. 4A, right and EV4A,B).

Because S-phase duration remained similar with nucleoside addition despite significant fork acceleration, we reasoned that the number of active origins must have decreased (Fig. 4B). IODs clearly correlated with fork speed as reported previously (Maya-Mendoza et al, 2018; Zhong et al, 2013): in MEFs, IODs were overall shorter in early S phase while longer in late S phase, while IODs were consistently shorter in ES cells when the early and late fractions were compared. With nucleoside addition, IODs increased with fork acceleration, while the addition of ATRi caused a reduction in the IODs (Fig. 3B). Thus, the balance between fork speed and origin firing rate can be altered such that the S-phase length remains roughly the same.

## Replication fork acceleration causes miscoordination between DNA replication and cell cycle progression in ES cells

Because nucleoside addition reduced the duration after PCNA disappearance and the onset of chromosome condensation, we considered the possibility that completion of genome-wide DNA replication and subsequent cell cycle progression is mis-coordinated by nucleoside supplementation. A recent report suggested a mechanism in which DNA replication activity is monitored via events linked to origin activation (Zonderland et al, 2022). Thus, a reduced number of active origins may reduce the sensitivity for ongoing replication and may lead to premature entry into G2/M phase especially in ES cells, where CDK activity is inherently high.

To test if cell cycle coordination is disturbed upon nucleoside addition, we pulse-labeled cells with EdU to mark replicating cells and stained for a phosphorylated form of histone H3 (H3Ser10Ph), a marker for G2/M stages. Nucleoside addition increased the cell population positive for both EdU and H3Ser10Ph in ES cells but not in MEFs (Fig. 4C,D). Thus, nucleoside addition causes entry

into G2/M phase prematurely before the completion of S phase in ES cells.

We then tested to see if ES cells experience under-replication with nucleoside addition. Under normal culturing conditions, ~20% cells form at least one ultra-fine bridge (UFB) through M phase in ES cells, but nucleoside supplementation increased this to ~40% (Fig. 4E,F). A pulse treatment with a CDK inhibitor reduced the rate of UFB frequency to ~20% even in the presence of nucleoside, strongly suggesting that the elevated UFB formation is due to the miscoordination between genome-wide DNA replication and cell cycle progression. Micronuclei formation occurred at a higher rate upon nucleoside addition in ES cells but not in MEFs (Fig. 4G; Appendix Fig. S1D). This is consistent with the fact that nucleoside addition does not increase the number of cells double-positive for EdU and H3Ser10Ph (Fig. 4D). Taken together, our data suggest a link between ES-specific replication fork dynamics and coordinated cell cycle progression in ES cells.

In summary, our study highlights the dynamic and flexible aspect of the DNA replication mechanism. In ES cells, in which keeping gap phases short is important for maintaining pluripotency and therefore maintains high level of CDK activities, replication origins are programmed to license at higher levels and to fire at higher frequencies throughout S phase (Fig. EV4C). Although nucleosides can be used to accelerate replication fork speed, ES do not present signs of fork pausing under the normal culturing condition. On the other hand, a low level of fork pausing is observed specifically in the early S phase of MEFs, which is reduced by nucleoside addition. Hence, in MEFs, there may be dNTP shortage during a small window in early S phase, which may lead to ATR activation. Our findings indicate that active origin firing is not a result of elevated replication stress, but rather is a primary cause of reduced fork rate. The higher number of replication forks maintained on the genome to the end of S phase may enable replication activities to be more visible to the surveillance system (Fig. EV4C). This aspect may distinguish pluripotent stem cells from similarly fast-cycling cells, such as cancer cells. For cancer cells, an active cell cycle with fork acceleration and reduction in origin firing may lead to further genome instability, while ES cells may tolerate this by maintaining a high number of active origins.

## Methods

### Cell culture

The mouse ES cell line, E14Tg2A (Hooper et al, 1987), was maintained in G-MEM (078-05525, WAKO) with 10% FBS

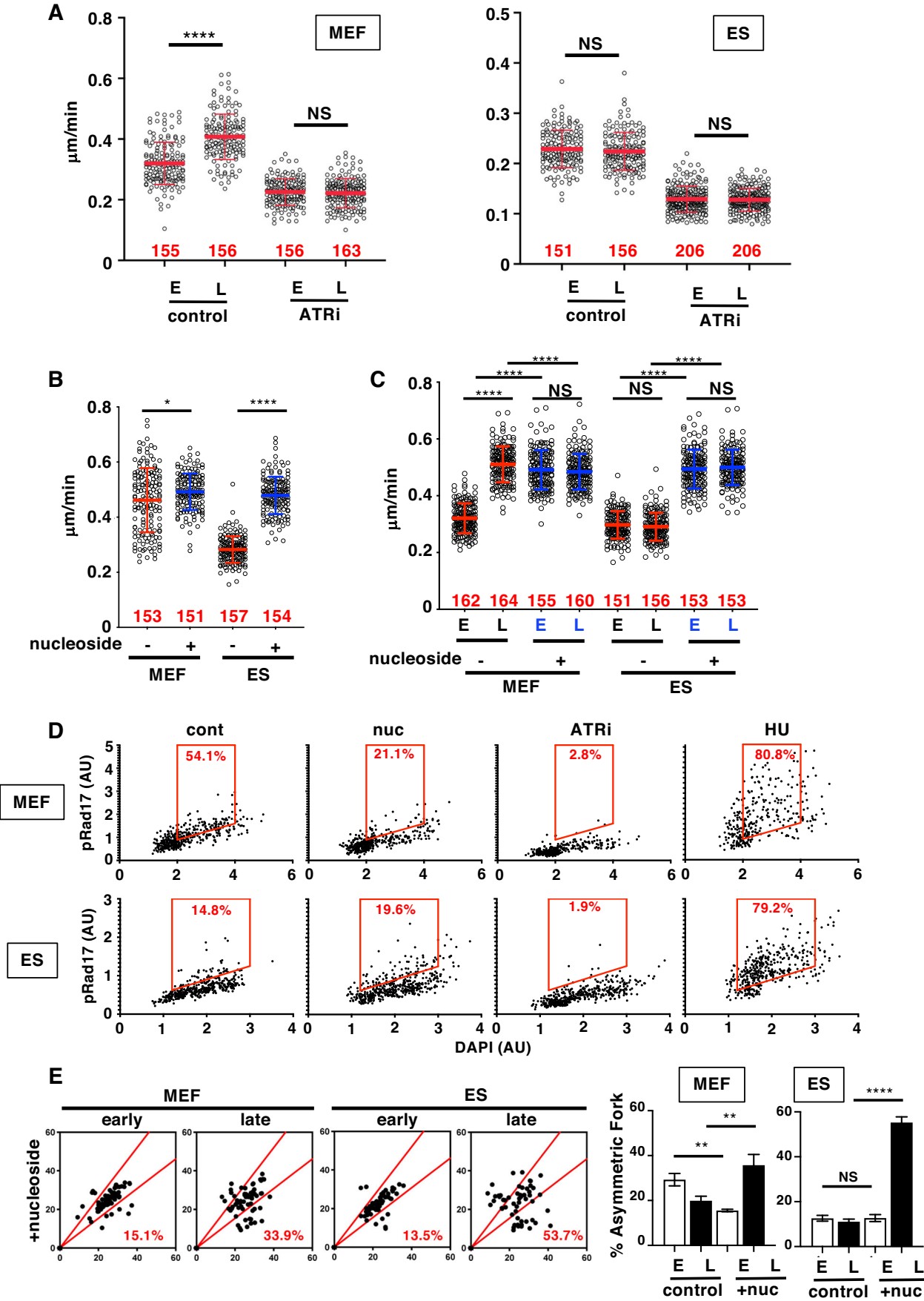

**Figure 3. Nucleoside supplementation leads to different cellular responses in MEF and ES cells.**

(A) Cells were treated with or without an ATR inhibitor (ATRi; 1 μM for 2 h) before fixation and FACS sorting. Cells were labeled with IdU and CldU during the final 30 min of the ATRi treatment. Sorted cells were then subjected to the DNA fiber assay. (B) Cells were treated with or without nucleosides for 48 h, sequentially pulse-labeled with IdU and CldU prior to fixation as in (A). (C) Cells collected as in (B) were fixed and sorted for early (S) and late (L) S-phase fraction as in Fig. 1B. (D) Cells were treated with or without nucleosides, ATRi (1 μM for 2 h), or HU (1 mM for 2 h), fixed and stained with anti-pRad17 and anti-PCNA antibodies and DAPI. Scatter plots of pRad17 intensity against DNA content (DAPI intensity) are shown. pRad17-positive populations within S phase (judged by PCNA staining) are gated with red boxes, and the percentage of the population among the total population is indicated in each box. (E) Fork asymmetry rates in early and late S fractions after 48 h of nucleoside incubation ($n = 3$). Data are presented as mean $+/-$ SD. NS, not significant ($P \geqq 0.05$), *$P < 0.05$, **$P < 0.01$, ****$P < 0.0001$, Mann–Whitney tests (A–C) and two-tailed $t$ test (E). The numbers of scored forks are indicated in red (A–C). Data from a representative experiment is shown ($n = 2$ independent experiments) for (A–D). (E) The mean $+/-$ SD of three independent experiments are shown (right) and their representative data are shown (left). Source data with exact $P$ values and measurements are available. Source data are available online for this figure.

(BioWest), MEM Non-Essential Amino Acids (NEAA, Gibco), 1 x Sodium Pyruvate (Gibco), 0.1 mM β-mercaptoethanol (WAKO), 1× Penicillin–Streptomycin (WAKO) and Leukemia Inhibitory Factor (LIF). The MEF cell line, SNL (McMahon and Bradley, 1990) was maintained in DMEM (D6546, Sigma-Aldrich) with 10% FBS, 1× NEAA, 2 mM L-Glutamine (WAKO), 0.1 mM β-mercaptoethanol, 1× Penicillin–Streptomycin. Neuro2A (N2A) cells (provided by Dr. Akihiro Fujikawa, NIBB) were cultured in DMEM supplemented with 10% heat-inactivated FBS and L-Glutamine. Abelson transformed mouse pre-B cells(Palmieri et al, 1994) are cultured in RPMI-1640 supplemented with 20% FCS, 1× NEAA, 2 mM L-Glutamine, 50 μM β-mercaptoethanol, antibiotics, and 5 ng/ml interleukin (IL)-7. RPE cells (provided by Atsushi Shibata, Keio University) were cultured in DMEM with 10% FCS, supplemented with 2 mM L-Glutamine and 1× Penicillin–Streptomycin. HeLa cells (provided by Atsushi Shibata, Keio University) were cultured in MEM (11090-081, Gibco) supplemented with 15% heat-inactivated FBS, 2 mM L-Glutamine, 1× Penicillin–Streptomycin. Human iPS cells (PChiPS771) were purchased from REPROCELL (Cat No. RCRP001N) and maintained in StemFit AK02N medium (REPROCELL) according to the manufacturer's protocol. All cell lines were cultured at 37 °C under 5% $CO_2$. To differentiate mouse ES cells, cells were cultured in media without LIF for three days. For nucleoside supplementation, EmbryoMax® Nucleosides 100× (ES-008-D, Sigma-Aldrich) were used with 50-fold dilution (2× the suggested concentration).

## DNA fiber assay

DNA fiber assay was carried out as described previously (Kawabata et al, 2011). Briefly, cells were sequentially labeled with 20 μM IdU (I7125, Sigma-Aldrich) and 200 μM CldU (C6891, Sigma-Aldrich) for 15 min each, unless otherwise indicated. The cells were then washed and resuspended in PBS, Ethanol (70%v/v), or Methanol (3:1 methanol/acetic acid) depending on the downstream procedure. To sort cells and measure fork dynamics in substages of S phase, Ethanol was used to enhance propidium iodide (PI) staining, but otherwise PBS or methanol were used. The methanol fixation is described previously (Kurashima et al, 2020). To stretch DNA fibers on slides, two microliters of fixed cells were deposited onto APS-coated slide glass and incubated for 2 min. Then, 7.5 μl of lysis buffer [0.2 M Tris-HCl (pH 7.5), 50 mM EDTA (pH 8.0), 0.5% SDS] was dropped onto the cells and incubated for 6 min. For EtOH-fixed cells, proteinase K (Invitrogen, 25530049) was supplemented to the lysis buffer at 0.4 μg/μl and incubated for 10 min. The DNA fibers released from the cells were extended by

tilting the slides in a high-humidity chamber over 30 min. The slides were immersed in fixative solution for 2 min and washed in distilled water. To denature the DNA fibers, the slides were immersed in 2.5 M HCl for 80 min and washed three times in PBS. After blocking with PBS containing 1% BSA for 20 min, the slides were incubated for 2 h at room temperature (RT) with anti-IdU (1/400, Sigma-Aldrich, SAB3701448) and anti-CldU (1/25, Abcam, ab6326) antibodies in CanGetSignal A (TOYOBO, NKB-501) to label nascent DNA. After three washes with PBST, the slides were fixed with 3.7% formaldehyde in PBST for 10 min, washed three times with PBST, and incubated at RT for 1 h with anti-rat IgG conjugated with Alexa Fluor 594 (Jackson ImmunoResearch, 712-585-153) and anti-mouse IgG conjugated with Alexa Fluor 488 in CanGetSignal A. The slides were washed three times in PBST and mounted with Vectashield Plus (Vector Laboratories, H-1900). Images were captured with a fluorescence microscope (Olympus) and analyzed using ImageJ software (National Institutes of Health). Where indicated, micrometers were converted to kilobase pairs by multiplying the number of micrometers by 3.5 kb. To score fork speed, DNA fibers displaying ~1:1 length ratio of IdU:CldU were used to score the length of the IdU-CldU labeled region and were presented as fork speed (μm/min). At least 150 forks were examined in each sample. Sister fork asymmetry was measured as the ratio of the lengths of the left- and right-moving sister forks. Asymmetric forks were defined as sister forks differing >30% in length. To score asymmetric forks, at least 50 forks were examined in each sample.

## Inter-origin distance (IOD) analysis

IODs were scored as described previously (Kawabata et al, 2011), using slides that were used to score fork speeds. DNA fibers displaying a CldU-IdU-CldU staining pattern were identified as initiation sites (see Appendix Fig. S1B), and those presenting two or more initiation sites side-by-side were identified. The distances between the two adjacent initiation sites (the center of the IdU-labeled region) were measured as IODs. At least 50 IODs are scored in each sample.

## Flow cytometry

Logarithmically growing cells were fixed for overnight or longer in 70% ethanol at −30 °C. After two washes in PBS, the cells were stained with 50 μg/ml propidium iodide (PI) (81845, Sigma-Aldrich) in the presence of 100 μg/ml RNase A (12091-021, Invitrogen) for 30 min at RT. Data were acquired and analyzed using a SONY Cell Analyzer SH800 or MA900. Sorting of cells at different substages of S phase were

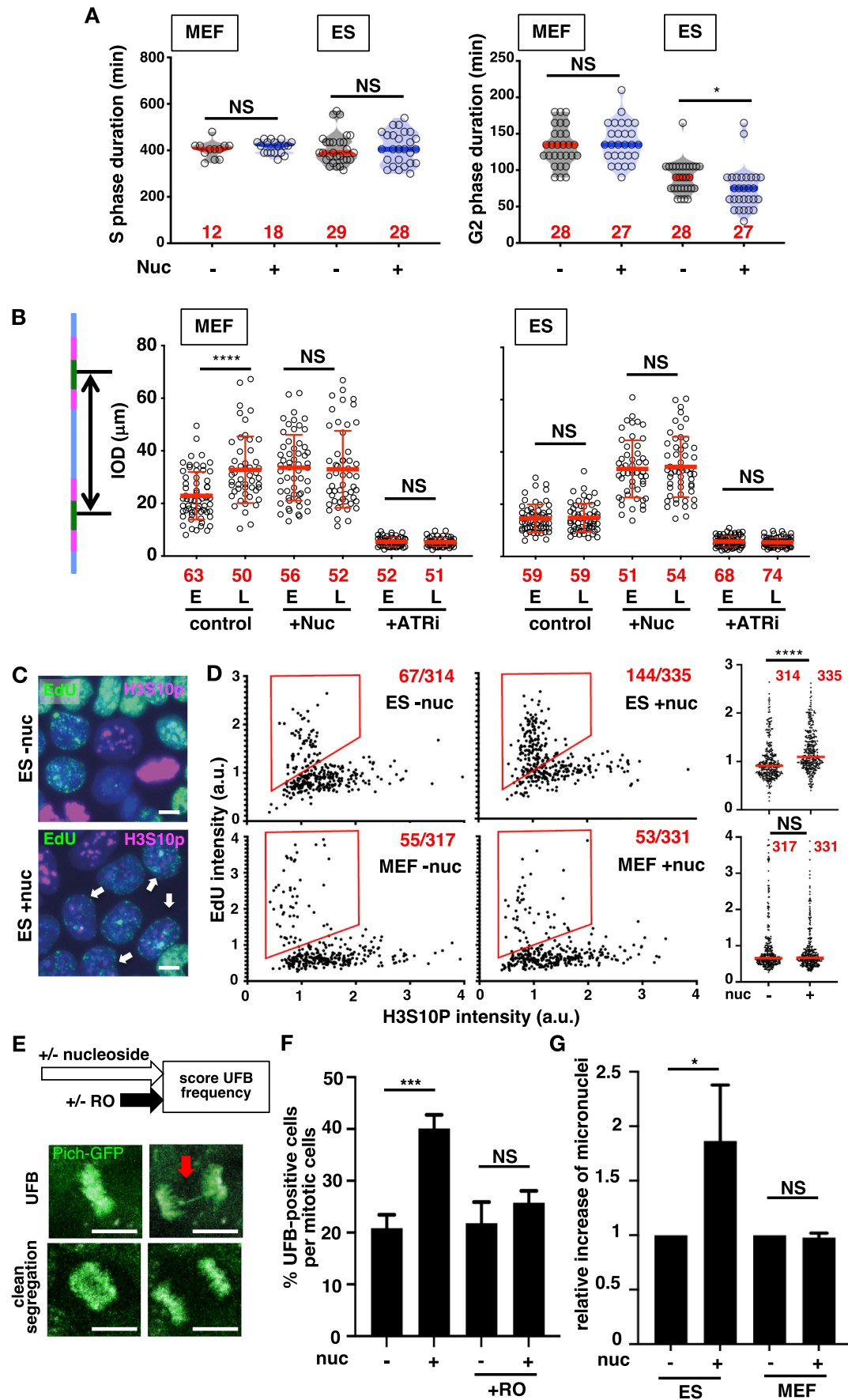

◀ **Figure 4. Nucleoside supplementation causes miscoordination between genome replication and cell cycle progression.**

(A) The duration of S- and G2- phases were measured based on time-lapse imaging using ES and MEF cells expressing PCNA-mNeonGreen. Cells were cultured with or without nucleosides for 24 h, and were subjected to imaging with 15-min intervals for 24 h. S-phase duration was defined as the duration when PCNA appears as foci (Fig. EV4A). The time from PCNA foci disappearance to the dispersal of the PCNA signal out to the cytoplasm (i.e., nuclear membrane breakdown) is scored as G2 phase duration. Bars indicating the median. The numbers of scored cells are indicated in red. Experiments were carried out in triplicate and are shown in Fig. EV4. (B) Inter-Origin Distances (IOD) were measured in cells treated with or without nucleosides or ATRi (1 μM for 2 h) and sorted. Data are presented as mean +/− SD. Data from a representative experiment is shown (n = 2 independent experiments). (C) Coordination between DNA replication and cell cycle progression was evaluated by co-staining with EdU and antibodies against histone H3 protein phosphorylated at Ser10 (H3S10P). Cells were pulse-labeled with 10 μM EdU for 15 min before fixation to label ongoing replication. White arrows indicate cells double-positive for EdU and H3S10P. Bar, 10 μm. (D) Only cells positive for H3S10P in (C) were extracted to plot for EdU and H3S10P intensities. Cells in the red box were scored as double-positive cells (Numbers of double-positives among H3S10P-positive cells are indicated in red). (right) Distribution of EdU intensities in the plotted (H3S10P-positive) population. Red bars indicate the mean. Data from a representative experiment is shown (n = 2 independent experiments). (E) The frequencies of ultra-fine bridge (UFB)-formation appearance with or without nucleosides were scored by time-lapse imaging. Nucleosides were added to the media 24 h prior to imaging, and cells were imaged for 12 h to capture cells undergoing mitosis. To evaluate the effect of transient cell cycle arrest in G2, cells were treated with a CDK inhibitor (RO) for 3 h (final concentration 10 μM) and washed out before imaging. The red arrow indicates examples of a UFB formed between segregating chromosomes. Bar, 10 μm. (F) Bar graph showing the percentage of UFB-positive cells per mitosis. Values represent the mean with SD (n = 3 independent experiments). In total, 32–114 mitoses were scored per condition. The exact numbers of mitoses scored for each condition are listed in the source data. (G) Bar graph showing relative frequencies of micronuclei appearance after 48 h of nucleoside treatment. In all, 1000–3000 of cells are scored. Values represent the mean with SD (n = 3 independent experiments). The exact numbers of cells scored are listed in source data. NS, not significant (P ≧ 0.05), *P < 0.05, ***P < 0.001, ****P < 0.0001, two-tailed t tests (A, D, F, G) and Mann–Whitney test (B). Source data with exact P values are available. Source data are available online for this figure.

**Table 1. Primers used for qPCR.**

|  | Sequence 5'–3' |
| --- | --- |
| Gapdh-F | TGCACCACCAACTGCTTAGC |
| Gapdh-R | GGCATGGACTGTGGTCATGAG |
| Esrrb-F | TTTCTGGAACCCATGGAGAG |
| Esrrb-R | AGCCAGCACCTCCTTCTACA |
| Sox2-F | CCATGGGCTCTGTGGTCAAG |
| Sox2-R | CCCTGGAGTGGGAGGAAGAG |
| Nanog-F | ACCTGAGCTATAAGCAGGTTAAGAC |
| Nanog-R | GTGCTGAGCCCTTCTGAATCAGAC |
| Pou5f1-F | CACGAGTGGAAAGCAACTCA |
| Pou5f1-R | AGATGGTGGTCTGGCTGAAC |
| Rex1-F | CTCCTAGCCGCCTAGATTTCCA |
| Rex1-R | CGTGTCCCAGCTCTTAGTCCATT |
| Klf4-F | TCCTTTCCAACTCGCTAACCC |
| Klf4-R | CGGATCGGATAGCTGAAGCTG |
| Sall4-F | CAATAGCCAAGCCGGAAGTGTC |
| Sall4-R | GTTGGAGGGAGGCTGGTACG |

Primers used in Fig. EV3B are listed.

done using gates indicated in Fig. 1 based on PI intensity. At least 5000 cells were sorted for each gate and were resuspended in ~100 μl PBS to carry out DNA fiber assay.

For γH2AX staining, cells were fixed with 4% paraformaldehyde in PBS for 15 min and permeabilized with Click-iT saponin-based permeabilization and wash reagent (Thermo Fisher Scientific, C10634) for 15 min at RT. The cells were then blocked with 3% BSA in PBS for 30 min and incubated for 2 h with primary antibodies against γH2AX (Millipore, 05-306, 1/100) in PBS containing 3% BSA and 0.05% Tween 20 at RT. After washing, cells were incubated with secondary antibodies conjugated to Alexa-488 (Jackson ImmunoResearch, 115-545-146) for 1 h at RT. DNA was stained with FxCycle Violet (Thermo Fisher Scientific, F10347).

## Generation of an mNeonGreen-PICH and mNeonGreen-PCNA stable cell lines

To visualize PICH and PCNA proteins during live imaging, an mNeonGreen cassette was inserted at the N-terminal end of the targeted gene at the endogenous locus using CRISPR/Cas9 system. In brief, a plasmid containing donor sequences (pGbait-Puro2A-mNG-Pich or pGbait-Puro2A-2xmSCl-Pcna) was co-transfected with a plasmid expressing Cas9 and gRNA (pX330S-sgPich and pX330S-sgPcna) with lipofectamine 3000 following the manufacturer's protocol. Detailed information regarding the plasmid construction is listed below. 24 h after transfections, cells were re-plated at different densities and were selected with 1 μg/ml Puromycin (160-23151, Wako) for 7–10 days before isolating clones. Insertions were confirmed by sequencing of the targeted region.

## Plasmids

For plasmids expressing guide RNA and Cas9 for gene editing, the sense and antisense oligonucleotides [for *Pich* (5'-TCCGGATTCCCTGGC CATGG-3') or *Pcna* (5'-AGCTACTCTAAGAGGAACG-3')] that target near the start codons were annealed and cloned into pX330S-2 (Addgene #58778) at gRNA scaffold sites (to obtain pX330S-sgPich and pX330S-sgPcna) using T4 DNA ligase. For donor plasmids (pGbait-Puro2A-mNG-Pich and pGbait-Puro2A-2xmSCl-Pcna), approximately 500 bps of upstream and 500 bps downstream of the Cas9-targeted site were amplified using the genomic DNA of E14Tg2A as a template and assembled with mNeonGreen cassette fused with puromycin N-acetyltransferase via P2A using NEBuilder HiFi DNA Assembly (E2621, New England Biolabs), following the manufacturer's protocol.

## Chemicals

The ATR inhibitor VE822 and the CDK1 inhibitor (RO-3306) were purchased from Selleck (S7102) and Calbiochem (217699) respectively. Hydroxyurea (HU) was purchased from TCI Chemicals (H0310), Cordycepin and N-Acetyl-cysteine (NAC) were

purchased from Wako (017-05131), 5,6-dichloro-1-β-d-ribofura-nosylbenzimidazole (DRB) from Cayman Chemical (10010302), and triptolide from AdipoGen (AG-CN2-0448). 5-Ethynyl Uridine (EU) was purchased from Thermo Fisher Scientific (E10345). CDC7 inhibitor PHA-767491 was purchased from Sigma-Aldrich (PZ0178).

Concentrations and conditions used in each assay are listed in Figure legends.

## Live imaging for cell cycle and UFB analyses

To measure cell cycle duration, mNeonGreen-PCNA expressing ES cells were treated with SirDNA (CY-SC007, Spirochrome) for 30 min to 1 h, before starting time-lapse-imaging using CellVoyager CV1000 (Yokogawa Electric) equipped with an EMCCD camera and either a 40× UPLSAPO40X2 (NA0.95) or 60X UPLSAPO60XS (NA1.3) objective. Seven z-stacks at a 2 μm interval (stack range 12 μm) were imaged every 15 min. S-phase duration was estimated as the period from the appearance to the disappearance of PCNA foci, and the duration from there to PCNA signal dispersal into the cytoplasm was defined as G2 phase. To quantitate frequencies of UFB appearance, mNeonGreen-PICH expressing ES cells were treated with SirDNA, and time-lapse-imaging was performed using CellVoyager CV1000. Six z-stacks at a 2.4 μm interval (stack range 12 μm) were captured every 90 s. UFBs formed between sister chromatids separated by 5 μm or more were considered positive. At least 30 mitoses and 7 UFBs were scored per experiment.

## Immunofluorescence assays

Cells underwent pre-extraction with 0.05% Triton X-100 in PBS for 5 min on ice, followed by fixation in 4% paraformaldehyde in PBS for 10 min at RT (for EdU, Oct4 and phosho-H3S10) or methanol for 20 min at −30 °C (for phospho-Rad17 and PCNA). After washing with PBS, cells were post-extracted in 0.25% Triton X-100 in PBS for 10 min at RT and subsequently blocked with 3% BSA in PBS. The cells were then incubated at RT for 1 h with primary antibodies against Oct4 (1/1000, BD Biosciences, 611202), phospho-H3S10 (1/1000, Cell Signaling Technology, 53348), phospho-Rad17 (S645) (1/1000, Bethyl Laboratories, A300-153A) or PCNA (1/1000, Cell Signaling Technology, 2586) in PBS containing 3% BSA and 0.05% Tween 20. After washing, cells were incubated with secondary antibodies conjugated to the appropriate fluorophore [Alexa-488 or Alexa-594 (1/1000, Jackson ImmunoResearch, 115-545-146 or 115-585-144)] for 1 h at RT followed by DAPI staining. Replication foci were visualized using the Click-iT Plus EdU Imaging kit (Thermo Fisher Scientific, C10632) according to the manufacturer's protocol. Immunofluor-escence images were obtained with IX83 (Olympus) equipped with an Olympus LUCPlan N 40X objective lens and sCMOS camera (Zyla 5.5, Andor). Signals of EdU, phospho-Rad17, PCNA, phospho-H3S10, and DAPI were quantified using the "Measure-ment and ROI tool" in the cellSens software (Olympus). The total intensity [(average intensity-background intensity) x area] of each nucleus was measured by setting ROI as a DAPI-stained area. The background signal intensity was determined by setting ROI in three random regions without cells and determining the average intensity.

## Nascent RNA detection

Cells were pulse-labeled with 1 mM Ethynyl uridine (EU) for 30 min prior to fixation. When transcription inhibitors were used, cells were treated with inhibitors for 2 h during which co-treatment with EU was done during the last final 30 min. Cells were fixed with 3.7% formaldehyde in PBS for 15 min at RT. EU detection was performed using the Click-iT RNA Imaging Kit (Thermo Fisher Scientific, C10329) following the manufacturer's instructions. Fluorescence images were captured and analyzed as described above.

## Real-time PCR

Total RNA was isolated using the RNeasy kit (Qiagen, 1026497) according to the manufacturer's instructions. Reverse transcription reactions were performed on 1.5 μg of total RNA with the ReverTra Ace qPCR kit (TOYOBO, FSQ-301). Real-time quantitative PCR was conducted using THUNDERBIRD Probe qPCR Mix (TOYOBO, QPX-201). Gene-specific primers are listed in Table 1. The results were analyzed with the software installed in LightCycler 96 (Roche).

## Western blot

Whole-cell lysates (WCL) were prepared by directly lysing cells in SDS buffer (0.125 M Tris-HCl, pH 6.8; 4% SDS; 10% sucrose; 0.01% bromophenol blue (BPB), 0.2 M DTT). For cytoskeleton (CSK) buffer-insoluble fractions, cells were extracted in CSK buffer [10 mM PIPES, 100 mM NaCl, 3 mM MgCl$_2$, 1 mM EGTA, 0.2% Triton-X, 300 mM sucrose and protease inhibitor cocktail (Roche)] on ice for 3 min. After two washes with PBS, SDS buffer was added and lysed. Lysates were incubated at 95 °C for 5 min. The protein concentrations in the samples were quantitated by the Bradford assay (XL-Bradford, Apro Science). Equal amounts of protein were resolved by SDS-PAGE and transferred to PVDF membranes (Millipore) at 150 mA for 16 h at 4 °C. The membranes were blocked by incubation with 5% skim milk in PBST (1× PBS with 0.1% Tween 20), and incubated with primary antibodies in 5% skim milk in PBST for 4 h at RT. The primary antibodies used are: anti-phospho-Rad17 (S645) (1/1000, Bethyl Laboratories, A300-153A) and anti-PCNA (1/1000, Abcam, ab133534). After three washes with TBST, the membranes were incubated with HRP-conjugated secondary antibodies (1/5000, Cell signaling technology, 7074 or 7076) in 5% skim milk for 1 h at RT. After four washes in PBST, the blots were developed using the ImmunoStar Zeta (Fuji Film) according to the manufacturer's instructions and imaged using a LAS 3000 luminescent image analyzer (Fuji Film).

## Statistical analyses

Results were compared using the two-tailed $t$ test, Kruskal–Wallis test or the Mann–Whitney test. All statistical analyses were performed using Prism 7 or 9 software (GraphPad).

# Data availability

This study includes no data deposited in external repositories.

The source data of this paper are collected in the following database record: biostudies:S-SCDT-10_1038-S44319-024-00207-5.

## Peer review information

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

## Acknowledgements

The authors would like to thank current and past members of the Tsubouchi lab for discussion and technical assistance, Trans-Omics Facility, Optics and Imaging Facility of NIBB Trans-Scale Biology Center, and ExCELLs for technical support. We would also like to thank Drs. Beth Rockmill and Tim Nelson for reading the manuscript and Dr. Hideo Tsubouchi (TITECH), Dr. Andrei Chabes (Umeå Univ), members of the Genome Damage and Stability Centre (Sussex Univ), especially Drs. Chris Chan, Jo Murray and Tony Carr for their discussion.

This work was supported by JSPS KAKENHI Grant Numbers JP16K07382, JP17H06477, JST PRESTO JPMJPR18K8, JST FOREST JPMJFR2008, and TAKEDA Life Science Research Grant to TT.

## Author contributions

**Kiminori Kurashima**: Data curation; Formal analysis; Validation; Investigation; Visualization; Methodology; Project administration; Writing—review and editing. **Yasunao Kamikawa**: Resources; Data curation; Formal analysis; Investigation; Visualization; Methodology; Project administration. **Tomomi Tsubouchi**: Conceptualization; Supervision; Funding acquisition; Validation; Investigation; Methodology; Writing—original draft; Project administration; Writing—review and editing.

Source data underlying figure panels in this paper may have individual authorship assigned. Where available, figure panel/source data authorship is listed in the following database record: biostudies:S-SCDT-10_1038-S44319-024-00207-5.

## Disclosure and competing interests statement

The authors declare no competing interests.

# Expanded View Figures

**Figure EV1.  Effects of culturing media, fixation procedures, transcription, and reactive oxygen species (ROS) on replication fork speed measured by DNA fiber assay.** ▶

(A) MEFs were grown in the usual (MEF) or ES media for 48 h and were subjected to DNA fiber assay. In both repeated experiments (exp1 and 2), the fork speed of MEFs in ES media was not slower compared to when they were grown in the usual media. (B) Transcription was inhibited with 50 μM cordycepin (COR), 25 μM dichloro-ribofuranosylbenzimidazole (DRB), and 1 μM triptolide (TPL) for 2.5 h in ES cells, and their effects were evaluated by EU-Click staining ("Methods", Appendix Fig. S1A). EU intensities per nucleus were scored and plotted. The numbers of scored cells are indicated in red. (C) Fork speeds in ES cells treated with transcription inhibitors as in (B). (D) The effect of reactive oxygen species (ROS) was assessed by adding a ROS scavenger, N-acetyl-l-cysteine (NAC). Low doses of HU are known to increase ROS levels. As expected, NAC reversed the fork slowdown observed with 10 μM HU, but NAC alone does not increase fork speed in ES cells, indicating that fork slowdown in ES cells is not due to ROS. (E) Fork speed measurements were done using ES cells that were collected after pulse-labeling and cell pellets were resuspended in PBS, 70% ethanol (EtOH) or 3:1 methanol/acetic acid (MeOH). Fixed cells were kept at 4 degrees overnight (PBS, MeOH) or at -30 degrees for 1day or 5 days (EtOH). (F) Replication fork speed of N2A, mB and RPE cells in early (E), early–mid (EM), mid–late (ML) and late (L) fractions (as done in Fig. 1D). (G) ES cells were either cultured in a normal condition (control) or without LIF for 3 days (-LIF) and were pelleted to carry out quantitative PCR to compare expression levels of genes associated with pluripotency (left). Gene expression levels were determined relative to *mGAPDH* which were then compared relative to control samples. (middle) Immunofluorescent staining for Oct4 and DNA (DAPI) in cells grown in normal media or media without LIF. (right) Cell morphology in control and -LIF conditions. Numbers of scored forks are indicated in red for (A, C–F). Data in (G) are presented as mean $+/-$ SD ($n = 3$ biological repeats, each with $n = 2$ technical repeats). NS, not significant ($P \geqq 0.05$), *$P < 0.05$, ***$P < 0.001$, ****$P < 0.0001$, Mann–Whitney tests (A, B, D), Kruskal–Wallis tests (C, E, F) and two-tailed *t* test (G). Exact *P* values are also indicated. Data from a representative experiment is shown ($n = 2$ independent experiments) for (A–F).

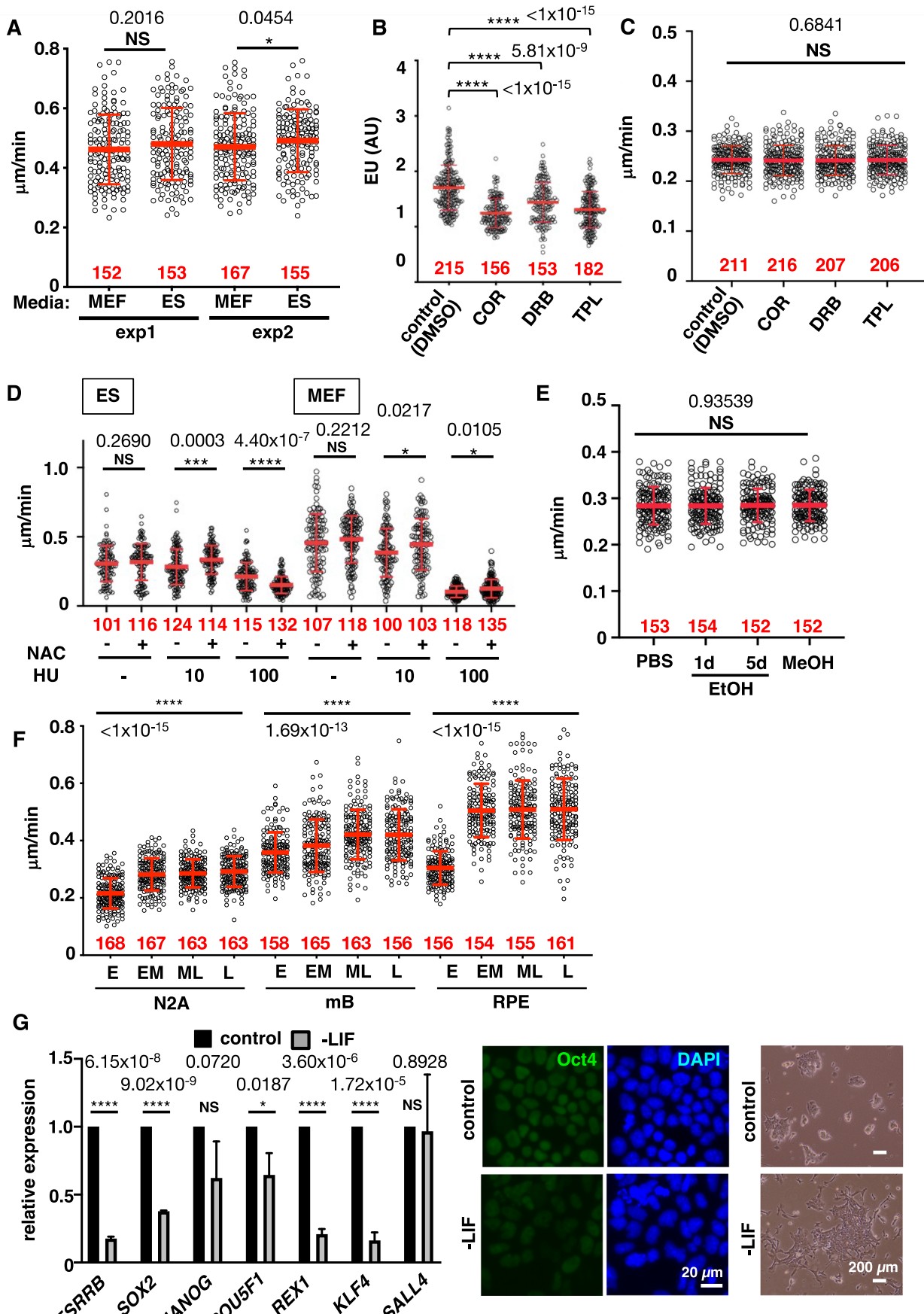

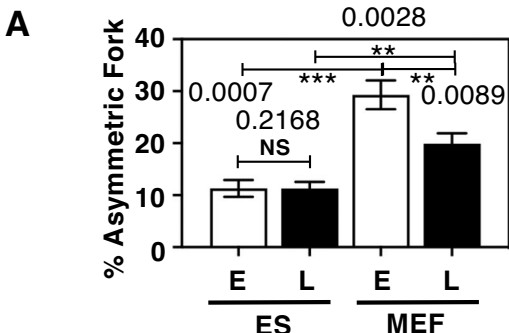

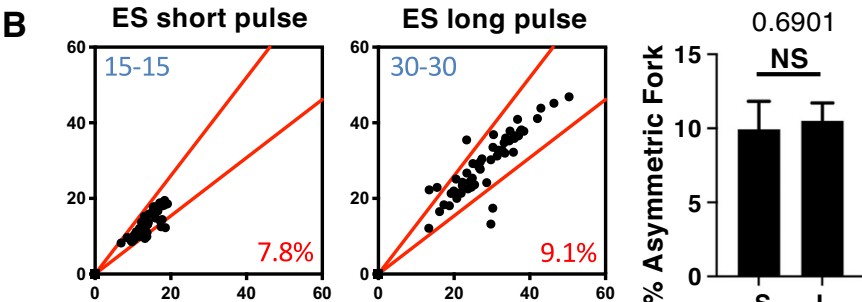

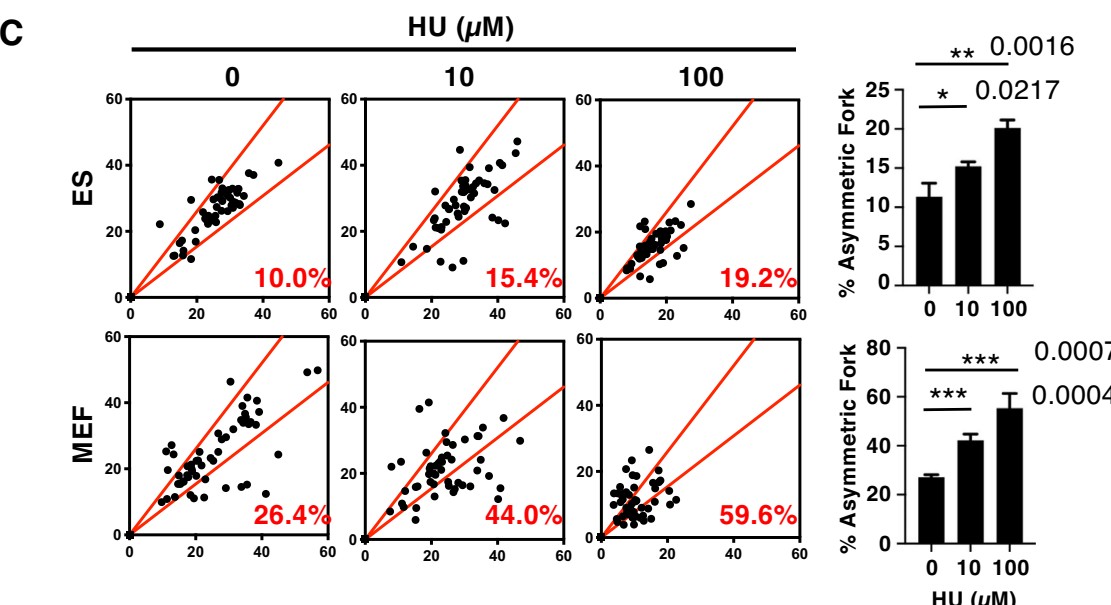

**Figure EV2. Effects of labeling time and HU on replication fork asymmetry.**

(A) Summary of fork asymmetry analyses shown in Fig. 2B ($n = 3$ biological replicates). Data are presented as mean $+/-$ SD of the % asymmetric forks (percentage of forks in the asymmetric region indicated in Fig. 2A) in the repeated experiments. (B) Fork asymmetry rate in ES cells with longer pulse-labeling times (IdU - 30 min and CldU - 30 min) ($= 30$-30, L) and compared with standard labeling (15–15, S). $n = 3$ independent experiments. Data are presented as mean $+/-$ SD. (C) Fork asymmetry rate in low (10 μM) and high (100 μM) concentrations of HU. $n = 3$ independent experiments. NS, not significant ($P \geqq 0.05$), *$P < 0.05$, **$P < 0.01$, two-tailed $t$ tests. Exact $P$ values are also indicated. At least 50 DNA fibers are scored to evaluate fork asymmetry rate per experiment.

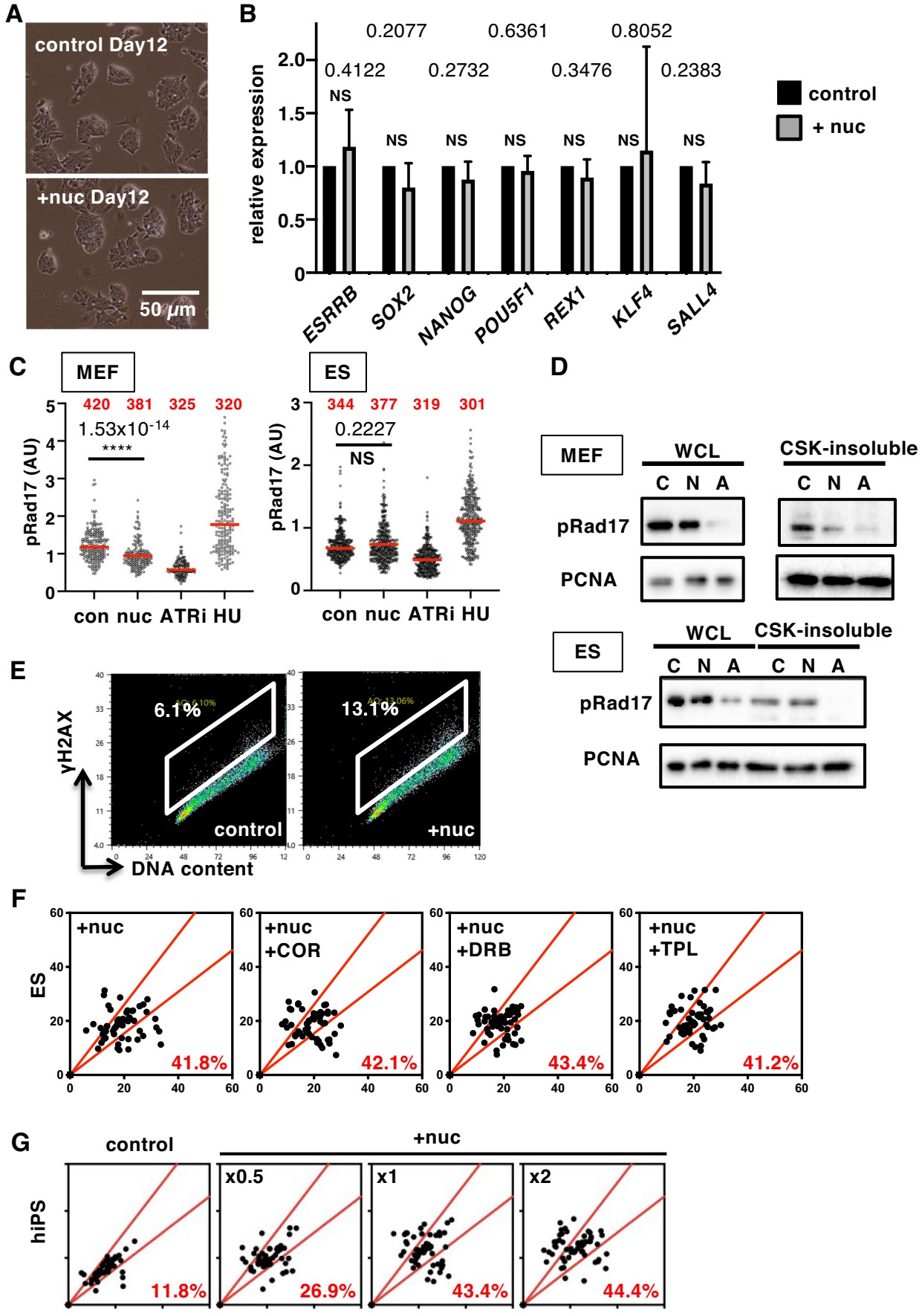

◀ **Figure EV3.  Effects of nucleoside addition.**

(A, B) ES cells were cultured with (+nuc) or without (control) nucleosides for 12 days with daily media changes, and their morphology (A) and pluripotency-associated gene expression (B: $n = 3$ biological replicates, each with 2 technical replicates) were evaluated. Data presented as mean $+/-$ SD. NS, not significant ($P \geqq 0.05$), two-tailed $t$ tests. (C) MEF and ES cells were treated with nucleosides (48 h), 1 μM ATRi (2 h) or 1 mM HU (2 h) prior to fixation and immuno-staining of the phosphorylated form of Rad17 (pRad17) was carried out as a measure of ATR activity. Signal intensities were quantitated for plots shown in Fig. 3D, but only within S phase which includes all the data points in between left and right borders (including points outside box, towards bottom). The number of cells scored are indicated in red. Red bars indicate mean. NS, not significant ($P \geqq 0.05$), ****$P < 0.0001$, Mann–Whitney tests. Exact $p$ values are also indicated. (D) MEF and ES cells were treated with (N) or without (C) nucleosides or 1 μM ATRi (A) for 2 h and were collected for western blot analysis. WCL; whole-cell lysate, CSK-insoluble; Cytoskeleton buffer-insoluble ($=$ nuclear fractions). PCNA is used as an internal control. (E) ES cells grown in normal condition (control) or with nucleosides for 48 h were fixed and stained with anti-γH2AX antibodies and a DNA binding dye (FxCycle Violet), and were subjected to flow cytometry. (F) Fork asymmetry analyses were carried out using ES cells treated with nucleosides for 48 h with or without transcription inhibitors (COR, DRB, TPL) for 2.5 h as in Fig. EV1B. (G) Human iPS cells were treated with different concentrations of nucleosides (0.5×, 1×, or 2× suggested concentrations) for 72 h and were collected for DNA fiber assay and fork asymmetry analyses. Data from a representative experiment is shown ($n = 2$ independent experiments) for (C–G).

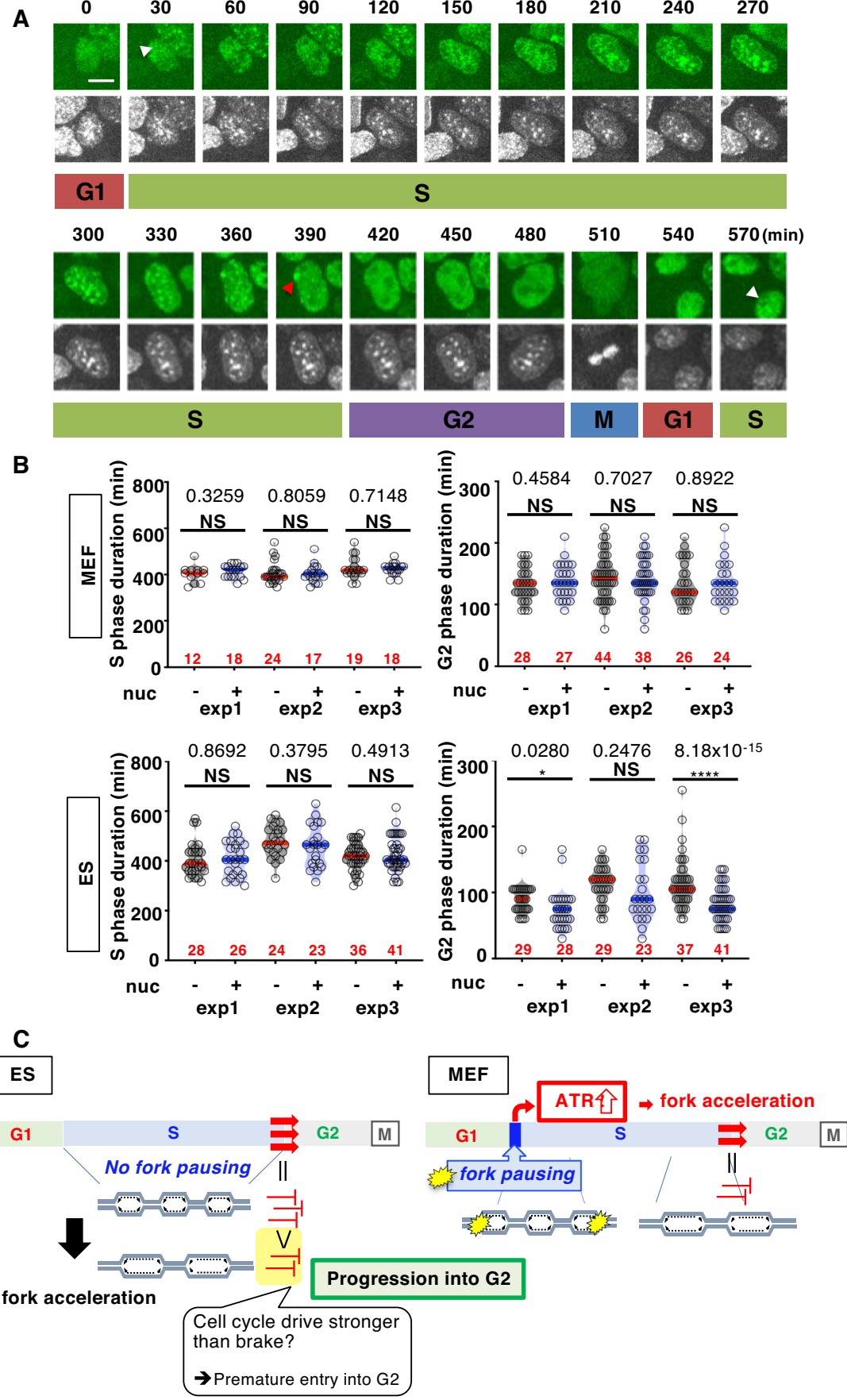

**Figure EV4.  Measurement of cell cycle duration and the model.**

(**A**) An example of time-lapse images of PCNA-mNG used in Fig. 4A and Fig. EV4B. The time at which the imaged frame exhibited PCNA foci (white arrowhead) was defined as the start of S phase and that of PCNA foci disappearance (red arrowhead being the last frame before disappearance) was defined as the end of S phase. The time at which PCNA signal spread out to the cytoplasm as a result of nuclear membrane breakdown, was defined as mitosis. Chromosome condensation is also evident at this point. Bar, 10 μm. (**B**) Repeat experiments shown in Fig. 4A. Exp1 is the same as the data shown in Fig. 4A. NS, not significant ($P \geqq 0.05$), *$P < 0.05$, ****$P < 0.0001$. Two-tailed *t* tests. Exact *P* values are also indicated. The numbers of scored cells are indicated in red. (**C**) A proposed model. See main text for details.

