## [Peer Review File · EMBO Reports]

Embryonic Stem Cells Maintain High Origin Activity and Slow Forks to Coordinate Replication with Cell Cycle Progression

Kiminori Kurashima, Yasunao Kamikawa, and Tomomi Tsubouchi

Corresponding author(s): Tomomi Tsubouchi (tsubo@nibb.ac.jp)

Review Timeline:

Submission Date:	25th Jan 24
Editorial Decision:	20th Feb 24
Revision Received:	13th May 24
Editorial Decision:	27th Jun 24
Revision Received:	1st Jul 24
Accepted:	3rd Jul 24

Editor: Esther Schnapp

Transaction Report:

Dear Dr. Tsubouchi,

Thank you for the submission of your manuscript to EMBO reports. We have now received the full set of referee reports as well as cross-comments from referee 2, which are all pasted below.

As you will see, the referees acknowledge that the findings are potentially interesting. However, they also point out that the study, as it stands now, does not provide a sufficient advance for publication here. Both referees 2 and 3 note that more insight into molecular mechanisms should be provided, which will be the case, if you can address all referee concerns. We can therefore invite you to revise your manuscript, if all concerns and suggestions can be addressed. Please let me know if you have any comments or questions regarding the referee reports, and we can discuss the exact revision requirements further, also in a video chat, if you like.

Assuming that you agree to address all concerns, I would like to invite you to revise your manuscript with the understanding that the referee concerns must be fully addressed and their suggestions taken on board. Please address all referee concerns in a complete point-by-point response. Acceptance of the manuscript will depend on a positive outcome of a second round of review. It is EMBO reports policy to allow a single round of major revision only and acceptance or rejection of the manuscript will therefore depend on the completeness of your responses included in the next, final version of the manuscript.

We realize that it is difficult to revise to a specific deadline. In the interest of protecting the conceptual advance provided by the work, we recommend a revision within 3 months (22nd May 2024). Please discuss the revision progress ahead of this time with the editor if you require more time to complete the revisions.

- 1) A data availability section providing access to data deposited in public databases is missing. If you have not deposited any data, please add a sentence to the data availability section that explains that.
- 2) Your manuscript contains statistics and error bars based on $n=2$. Please use scatter blots in these cases. No statistics should be calculated if $n=2$.

5) a complete author checklist, which you can download from our author guidelines

<<https://www.embopress.org/page/journal/14693178/authorguide>>. Please insert information in the checklist that is also reflected in the manuscript. The completed author checklist will also be part of the RPF.

6) Please note that all corresponding authors are required to supply an ORCID ID for their name upon submission of a revised manuscript (<<https://orcid.org/>>). Please find instructions on how to link your ORCID ID to your account in our manuscript tracking system in our Author guidelines

<<https://www.embopress.org/page/journal/14693178/authorguide#authorshipguidelines>>

10) Regarding data quantification (see Figure Legends:

<https://www.embopress.org/page/journal/14693178/authorguide#figureformat>)

- the name of the statistical test used to generate error bars and P values,

- the number (n) of independent experiments (please specify technical or biological replicates) underlying each data point,

- the nature of the bars and error bars (s.d., s.e.m.),

- If the data are obtained from n Program fragment delivered error ``Can't locate object method "less" via package "than" (perhaps you forgot to load "than"?) at //ejpvfs23/sites23b/embor_www/letters/embor_decision_revise_and_review.txt line 56.' 2, use scatter blots showing the individual data points.

You are able to opt out of this by letting the editorial office know (emboreports@embo.org). If you do opt out, the Review Process File link will point to the following statement: "No Review Process File is available with this article, as the authors have

chosen not to make the review process public in this case."

I look forward to seeing a revised form of your manuscript when it is ready.

Kind regards,
Esther

Referee #1:

Kurashima and his team present a manuscript aiming to validate the hypothesis that embryonic stem (ES) cells exhibit a distinctive DNA replication mechanism tolerant to genome instability. Their analysis of replication forks' dynamics and integrity led to the conclusion that ES cells maintain a slow fork speed throughout the S phase with a high density of active origins, avoiding fork stalling. In contrast, non-pluripotent cells demonstrate slow fork progression early in the S phase, some fork pausing, and later an acceleration in fork speed with reduced stalling. While intriguing, I find that the study contributes minimally to existing knowledge. Notably, Nakatani et al.'s Nature Genetics publication (Volume 54, pages 318-327, 2022) already provides excellent data on a similar topic. Beyond novelty, my concern extends to the experimental design, particularly the seemingly inadequately controlled cell sorting strategy for enriching cells in different S phase stages.

Established protocols for studying DNA replication timing, notably those from David Gilbert's laboratories, emphasize meticulous cell sorting. It appears that what Kurashima and colleagues label as the early S phase may actually include G1 cells, with only a few entering the S phase. Furthermore, what they identify as the late S phase may be cells in G2, as indicated in Figure 1b. These issues, among others, have dampened my enthusiasm to endorse the publication of this manuscript in a prestigious journal like Embo Reports.

Additional specific comments include the need for the authors to specify the exact number of scored forks, mean, and standard deviation values for each DNA fiber experiment. Considering the variation observed in Figure 1f and control treatments in Figure 3c, scoring only 150 forks may be insufficient, especially when compared with other publications. The authors primarily utilize the ES cell line E14tg2A for replication dynamics comparison with the MEF cell line, NLS. A suggestion is to induce the differentiation of E14tg2A to experimentally test some of their claims.

In general, the manuscript is well-written, but thorough proofreading is essential before submitting it elsewhere for publication.

Referee #2:

In this study, Kurashima et al compare DNA replication dynamics in mouse embryonic stem (ES) cells and differentiated mouse embryo fibroblasts (MEFs). The starting point is the known fact that, on average, replication forks are slower in ES than in MEFs (Fig 1a) and this leads to the observation that fork speed increases along S phase in MEFs, but it remains consistently low in ES (Figure 1b-f). Fork asymmetry, a surrogate measure for fork pausing events, is higher in MEFs than in ES, but the latter increases when early-S ES cells are treated with HU (Figure 2). Inhibition of ATR kinase slows down forks in both cell types, eliminating the difference between early- and late-S in MEFs (Figure 3a). Addition of nucleosides, previously known to increase fork speed in other cell types, results in the expected acceleration of ES forks and is accompanied by a marked increase in fork asymmetry, particularly in late-S phase (Fig 3b-e). As the duration of S phase is not affected by the addition of extra nucleosides, the authors report that faster fork progression is compensated by a decrease in the number of active origins (Figure 4b). Finally, the study shows that artificial acceleration of ES forks with nucleosides uncouples S phase and mitosis, leading to anaphase ultrafine bridges and formation of micronuclei (Figure 4c-g).

The regulation of DNA replication in cells with different pluripotency potential is a very interesting topic. Changes in fork speed and frequency of fork reversal in ES cells have already been reported, but the causes and mechanisms underlying these processes require more investigation. This manuscript by Kurashima and coworkers is clearly written and presents some new interesting observations, but in its current form it feels short of a significant advance, in my opinion. My recommendation at this point would be for the authors to gain more mechanistic insights before the manuscript is reconsidered for publication.

MAJOR POINTS

1. The result shown in Figure 3a (slowdown of fork progression by ATRi) does not take into account that ATR levels could be

- different between early and late S phase. This point could be addressed using the analysis of pRad17 as in Fig 3d and Supp 3d.
2. The same result (Figure 3a) would benefit much from a parallel analysis of origin activity, as done in Fig 4b.
 3. Fig 3d-e shows that ES forks forced to proliferate faster with nucleosides become asymmetric, i.e. display many more pausing events. A plausible explanation for this phenotype is a higher frequency of replication-transcription conflicts, and this should be addressed directly. At a minimum, the authors should test the transcription inhibitors used for the initial controls in Supp Figure 1. Other possibilities would be to estimate the frequency of R-loops, or to monitor RTCs by proximity-ligation assays of BrdU and RNA polymerase II.
 4. It would be useful to determine whether addition of extra nucleosides induces DNA damage markers (e.g. 53BP1 and gH2AX foci).
 5. The analysis of inter-origin distance (Figure 4b) is not described in Materials and Methods. In the methodology indicated for other DNA fiber labeling experiments, no antibody is used to monitor the integrity of individual DNA fibers. If this is the case, how can the authors be sure that two origin signals belong to the same fiber? These aspects should be clarified and the Methods section should be completed.
 6. The position of the gates in the flow cytometry experiments in Figure 4c-d is unclear. Double-positive cells should be in the top-right position, but the gates are in the top-left position, which are suggestive of EdU-positive only cells.
 7. As shown in Figure 4b, faster fork progression is compensated by a decrease in the number of active origins. The manuscript (p.8, l. 21) refers to this result as "Striking" but actually it is the expected result (the compensations between fork speed and origin activity are well established, as acknowledged by the authors). Also referring to this result, the authors state in the Discussion, "Whether fast replication limits the number of active origins or increased number of active origins slows down replication remains unclear at this point". This issue can be addressed experimentally with a relatively simple approach. Authors could check out a JBC paper about this topic (PMID: 29959228).

MINOR POINTS

1. The statistical analysis in Figure 3c should be expanded to include other comparisons like Early (-/+ nuc), or Late (-/+ nuc).
2. Discussion could mention recent progress in the field of DNA replication in early embryonic stem cells from the Torres-Padilla lab in Munich (e.g. PMID 35256805; PMID 38123678).
3. p.6, lines 5-6, sentence is incomplete.
4. p. 10, l. 10, typo in "endonegous"
5. Some references are incomplete or in the wrong format, e.g. refs 30, 34, 40.
6. Typo in supplementary Table title.

Referee #3:

The article "Pluripotent Stem Cells Maintain High Number of Replication Machinery to Coordinate with Cell Cycle" by Kurashima et al., explores the phenomenon of slow replication fork progression in pluripotent stem cells. The authors meticulously compare the regulation of fork speed in non-pluripotent cells to that of pluripotent ones, and describe specific differences, including increase of the average fork speed as non-pluripotent cells go from early to late S-phase, that is absent in the pluripotent cells. The authors also conclude that limiting fork speed is important to avoid genome instability, as artificially speeding it up by nucleoside supplementation resulted in UFB and micronuclei formation. Overall, the article is very well written and will be an important contribution to the field, and of interest to the broader readership of EMBO Reports. I believe it should be published, and I really only have a couple of comments below.

- 1) The title of the article does not match the content. There is nothing in the paper about the high number of replication machinery, or any experiments attempting to quantify it.
- 2) One question that remains unanswered is what mechanism limits fork speed in pluripotent cells, or what mechanism speeds up the forks in the non-pluripotent cells, that is absent in the pluripotent cells. Do they have lower nucleoside concentrations due to downregulation of one or more metabolic enzymes? It would be very interesting to compare nucleoside concentrations between the types of cells used in this study and whether they change during S-phase progression. Or maybe the authors have a different hypothesis on the mechanism of the fork speed regulation. If it is impossible within the timeline of revisions, this

should be discussed with the information available from the literature, and a brief schematic of the proposed model could be helpful.

Cross-comments from referee 2:

Regarding reviewer 1: I do agree that in its current form, the manuscript does not provide a major advance over previous knowledge (hence my set of suggestions). On the other hand, I would like to comment on the criticism over the cell sorting strategy used in Figure 1. It is true that the very early S phase fraction is likely contaminated with G1 cells, and the very late S is likely contaminated with G2 cells, and this could affect replication timing experiments of the type done by DM Gilbert and others. However, I do not think that this is a serious concern in the stretched DNA fiber approach, because the DNA from cells outside S phase cannot be labeled by CldU-IdU and therefore does not interfere with the measurements of fork speed or origin activity.

Regarding reviewer 3, point 2: I agree that comparing the nucleoside concentration of pluripotent vs mESCs would be interesting but it also may be a difficult task, as few laboratories have the biochemical tools to do it reliably. I agree that a better discussion of the possible reasons underlying the different fork speed would be useful.

Response to referees:

We are very grateful to the editor and referees for their constructive suggestions and criticisms, which we found extremely helpful improving our manuscript. Below, the original comments by the reviewers are in blue; cross-comments are in green. Changes in response to referees' comments have been made in red throughout the manuscript.

Comments to the Author

Referee #1:

..... While intriguing, I find that the study contributes minimally to existing knowledge. Notably, Nakatani et al.'s Nature Genetics publication (Volume 54, pages 318-327, 2022) already provides excellent data on a similar topic.

The same point was raised by referee #3 and we agree that this reference should have been cited properly. As these referees are aware, Nakatani et al. have investigated replication fork dynamics in the totipotent-like 2-cell-like cells (2CLCs) and in the corresponding early embryos. Interestingly, these cells present much slower replication forks compared to ES cells and they beautifully link replication fork dynamics to cellular plasticity. While I am aware that many people are reminded of their work with our data, our work is distinct from their work on several points. First, I would like to stress that embryonic stem (ES) cells and totipotent cells are completely different and distinct from each other. Considering the embryonic stages *in vivo* from which they are derived from, one has to remember that 2- and 4-cell stage embryos do not present transcription activity (zygotic gene activation begin between 4- and 8- cell stages of development), which is critically important when thinking about DNA replication dynamics. While the area is currently the subject of active investigation, chromosomal architecture of the embryos in the very early stages are likely to be very different from blastocysts, which also have much influence on replication dynamics. Second, we focus on replication dynamics in the context of genome stability rather than cellular plasticity. Technically, we present here meticulous documentation of replication fork dynamics in substages of S phase comparing pluripotent- and non-pluripotent cells, which have not been carried out to date. Although we did not have enough space to spell out all these points, we have included main points in the manuscript.

Beyond novelty, my concern extends to the experimental design, particularly the seemingly inadequately controlled cell sorting strategy for enriching cells in different S phase stages. Established protocols for studying DNA replication timing, notably those from David Gilbert's laboratories, emphasize meticulous cell sorting. It appears that what Kurashima and colleagues label as the early S phase may actually include G1 cells, with only a few entering the S phase. Furthermore, what they identify as the late S phase may be cells in G2, as indicated in Figure 1b.

Cross-comments from referee 2:

Regarding reviewer 1: I do agree that in its current form, the manuscript does not provide a major advance over previous knowledge (hence my set of suggestions). On the other hand, I would like to comment on the criticism over the cell sorting strategy used in Figure 1. It is true that the very early S phase fraction is likely contaminated with G1 cells, and the very late S is likely contaminated with G2 cells, and this could affect replication timing experiments of the type done by DM Gilbert and others. However, I do not think that this is a serious concern in the stretched DNA fiber approach, because the DNA from cells outside S phase cannot be labeled by CldU-IdU and therefore does not interfere with the measurements of fork speed or origin activity.

We apologize for not describing this point clearly in the manuscript. Regarding the concern raised by referee 1, how referee 2 has kindly described is exactly how we see our experiments. In theory percentage of cells in S phase in the first and last fractions may vary in different cell lines and experiments, but this would not change our conclusions.

Additional specific comments include the need for the authors to specify the exact number of scored

forks, mean, and standard deviation values for each DNA fiber experiment.

Per request, we have added the number of scored forks in the main figure and also documented mean, standard deviation values as a table in all figures and included as the source data.

Considering the variation observed in Figure 1f and control treatments in Figure 3c, scoring only 150 forks may be insufficient, especially when compared with other publications.

From our experience we feel that scoring of 150 forks are usually sufficient to deduce a reasonable conclusion. In fact, many papers, including the Nakatani paper discussed above, measures only ~50 forks in some cases (see Fig 1f). However, we have never carefully assessed this point. To gain clearer idea, we decided to collect the fork speed data for differentiated ES cells (per this referee's comment below) up to >300 forks and compared p-values in data sets with different number of forks collected (below).

As shown, overall trend does not significantly change across data sets. For comparison between ES-LIF E vs ES-LIF L, p-values were <0.0001 in all data sets and thus would not change our conclusion regardless of how many forks (within the range of 50-300) we score. However, p-values comparing ES-Con E vs ES-Con L would have given different impression: n=50-100 would give marginally significant difference ($p < 0.05$) while $n > 150$ would give no statistical difference ($p > 0.05$). In the originally submitted version of Fig1F, some data sets were short of 150 forks, so we have now re-scored all data. Comparison between differentiated vs undifferentiated ES cells scoring >300 forks are now presented in Fig1G.

The authors primarily utilize the ES cell line E14tg2A for replication dynamics comparison with the MEF cell line, SNL. A suggestion is to induce the differentiation of E14tg2A to experimentally test some of their claims.

Thank you for the suggestion. We indeed felt the need to test this point so we have carried out differentiation protocol employed by Ahuja et al (2016) of the Lopes group. By removing LIF for 3 days from the media we have been able to effectively differentiate mouse ES cells, leading to significant changes in morphology and *Oct4* expression at the protein level. Among 7 (*Esrrb*, *Sox2*, *Nanog*, *Oct4*, *Rex1*, *Klf4*, *Sall4*) genes that are generally used as pluripotency-associated markers, 5 were significantly reduced (Fig.EV1g). In this condition (indicated as ES-Diff in Figure 1G), replication fork speed dynamics drastically change and presented fork dynamics very close to MEFs. These results strongly suggest that slow replication fork dynamics throughout S phase is a feature linked to pluripotency.

In general, the manuscript is well-written, but thorough proofreading is essential before submitting it elsewhere for publication.

Our manuscript has now been proofread by two native speakers.

Referee #2:

.....The regulation of DNA replication in cells with different pluripotency potential is a very interesting topic. Changes in fork speed and frequency of fork reversal in ES cells have already been reported, but the causes and mechanisms underlying these processes require more investigation. This manuscript by Kurashima and coworkers is clearly written and presents some new interesting observations, but in its current form it feels short of a significant advance, in my opinion. My recommendation at this point would be for the authors to gain more mechanistic insights before the manuscript is reconsidered for publication.

We thank this referee for many constructive suggestions and we have now carried out the following experiments to gain more mechanistic insights. In particular, through experiments suggested by this referee in the following MAJOR POINT #5, we are now able to clearly show that the primary cause of fork slowing in ES cells is through origin activity rather than replication stress. We have also added data to show that 1) Differentiated ES cells behave similarly to non-pluripotent cells as shown here in that replication forks speed-up in late S phase 2) Nucleoside addition increases the level of a DNA damage marker γ H2AX (as expected and also previously shown by Ahuja et al using a mouse ES line). 3) The cause of DNA damage signaling is not through transcription activity. Along with these data sets, we have carried out all the experiments requested by the referees, increased data points and statistical data to strengthen our points.

MAJOR POINTS

1. The result shown in Figure 3a (slowdown of fork progression by ATRi) does not take into account that ATR levels could be different between early and late S phase. This point could be addressed using the analysis of pRad17 as in Fig 3d and Supp 3d.

Thank you for the suggestion. We have carried out measurements but were not sure how we can interpret the data. Therefore, the data was not included in the manuscript. At the face value, MEF cells present slightly higher ATR activity levels when evaluated with the amount of pRad17 staining levels (below). However, because ATR can be activated by many reasons and can have different downstream impact depending on the context, we are uncertain how we should interpret these results. What we think is safe to say, is that with ATR inhibition, fork acceleration does not take place in MEFs. We can also conclude that ATR is not absent nor inactive in ES cells, as ATRi affects overall fork speeds as expected from previous studies in different cell lines and as shown in MEFs in this study.

pRad17

2. The same result (Figure 3a) would benefit much from a parallel analysis of origin activity, as done in Fig 4b.

We have now added IOD analysis with ATRi and added this information in Fig4B.

3. *Fig 3d-e shows that ES forks forced to proliferate faster with nucleosides become asymmetric, i.e. display many more pausing events. A plausible explanation for this phenotype is a higher frequency of replication-transcription conflicts, and this should be addressed directly. At a minimum, the authors should test the transcription inhibitors used for the initial controls in Supp Figure 1. Other possibilities would be to estimate the frequency of R-loops, or to monitor RTCs by proximity-ligation assays of BrdU and RNA polymerase II.*

Thank you for your suggestion. To test whether the fork asymmetry observed upon nucleoside addition is caused by replication-transcription conflicts, we added transcription inhibitors in addition to nucleosides as suggested by the referee, and scored fork asymmetry rates. As shown in Figure EV3F, transcription inhibitors did not suppress fork asymmetry caused by nucleosides. Thus, it is unlikely that replication-transcription conflicts being the cause for fork asymmetry observed upon fork acceleration. We have now added this new information in the manuscript.

4. *It would be useful to determine whether addition of extra nucleosides induces DNA damage markers (e.g. 53BP1 and gH2AX foci).*

This point has actually been investigated by Ahuja et al. (2016) and are reported in Supplementary Fig. 5f of their paper. In their study they were seeking for the cause of DNA damage response activity and nucleosides were one of the conditions tested. They found that gH2AX levels were slightly increased using FACS in the presence of nucleosides, which we have also reproduced and included in the Figure EV3E. We have actually tried antibodies for other DNA damage markers including 53BP1 and Rad51 but the antibodies but were unable to quantitate the data due to poor staining quality.

5. *The analysis of inter-origin distance (Figure 4b) is not described in Materials and Methods. In the methodology indicated for other DNA fiber labeling experiments, no antibody is used to monitor the integrity of individual DNA fibers. If this is the case, how can the authors be sure that two origin signals belong to the same fiber? These aspects should be clarified and the Methods section should be completed.*

We sincerely apologize for not describing the inter-origin distance analysis in the methods section properly. As this referee points out, many studies employ DNA staining dyes to make sure that the origins being scored are on the same DNA fiber. However, if we do this for our sample, too many DNA fibers including the ones that are negative for CldU and IdU would be stained, thus making scoring extremely difficult. Instead, we follow the procedure employed by Kawabata et al. (Mol Cell 2011 PMID: 21362550), where they score IODs using DNA fiber samples that are stained for DNA analogs only. As shown below and in Appendix Fig. S1B, we can identify DNA fibers that are reasonably long even if there are short gaps within them. To test if we are missing longer IODs (for NOT staining DNA), we carried out DNA combing assay where we embed cells in agarose plugs and stretch DNA on slides to align DNA fibers better on slides and so we can stain DNA. DNA combing allows more stretching of the DNA, giving overall longer labeled DNA regions (in μm) with the same duration of pulse-labeling. When we normalized IODs so that we can compare distribution (using the stretching factor that we derived by comparing average $\mu\text{m}/\text{min}$ in the two assays), we did see that combing assay (with DNA staining) includes longer IODs. However, the bulk of IODs remains in the similar range when comparing with our data. Therefore, we are confident that our conclusions do not change with or without DNA stain.

From Kawabata et al. Mol Cell 2019 Figure1

6. The position of the gates in the flow cytometry experiments in Figure 4c-d is unclear. Double-positive cells should be in the topright position, but the gates are in the top-left position, which are suggestive of EdU-positive only cells.

We apologize for causing confusion. In this Figure (Fig3d), only H3S10P-positive populations were plotted (otherwise the plot would be massively dominated by EdU-positive H3S10P-negative population and would be difficult to see). Therefore, in this plot, EdU positive populations that are also positive for H3S10P-positive populations are shown. As they are mostly in population with low H3S10P intensities, they will appear at the left top corner of the overall population, rather than being on the right-top. H3S10P-positive populations with high intensity are generally in mitosis with condensed chromosomes, and EdU signals were not observed in these cells. We have now added the information in the figure legend.

7. As shown in Figure 4b, faster fork progression is compensated by a decrease in the number of active origins. The manuscript (p.8, l. 21) refers to this result as "Striking" but actually it is the expected result (the compensations between fork speed and origin activity are well established, as acknowledged by the authors).

We agree that the word "Striking" was not properly used in this sentence. We meant to point out that the result was very clear, but it was not properly stated. The sentence is now corrected (p9 line 3-4).

Also referring to this result, the authors state in the Discussion, "Whether fast replication limits the number of active origins or increased number of active origins slows down replication remains unclear at this point". This issue can be addressed experimentally with a relatively simple approach. Authors could check out a JBC paper about this topic (PMID: 29959228).

We thank this reviewer so much for this input as we have been struggling to decipher this question for long time and were not aware of this particular reference. In the mentioned JBC paper (PMID: 29959228), Rodriguez-Acebes et al. have taken advantage of a CDC7 inhibitor to

distinguish the cause of fork slowing and excess origin firing being primarily due to replication stress on fork progression or primarily on origin firing. If fork slowing is primarily caused by the replication stress on fork progression, suppression of origin-firing activity with the CDC7 inhibitor should not increase fork speed. On the other hand, if the origin activity being the primary cause of fork slowing, then CDC7 inhibition should increase replication fork speed. As stated in the original version of the manuscript, we did not believe that the mouse ES cells DO NOT experience significant levels of replication stress, hence the use of CDC7 inhibitor should increase replication fork speed. As expected, exposure of mouse ES cells to 10uM (final) CDC7 inhibitor for 2 hours significantly increased replication fork speed rate, indicating that the primary cause of fork slowing + higher number of active origins in ES cells is primarily caused by origin activity rather than increased levels of replication stress. Thus, we are now more confident to conclude that slow forks are not a result of replication stress but rather a secondary effect caused by (compared to non-pluripotent cells) excess origin firing. We have now added these discussions in the revised manuscript (p17).

MINOR POINTS

1. *The statistical analysis in Figure 3c should be expanded to include other comparisons like Early (-/+ nuc), or Late (-/+ nuc).*

We have now added -/+ nuc comparisons to Figure 3C. In addition to fork speeds, we also added -/+ nuc comparisons of fork asymmetry in Figure 3E.

2. *Discussion could mention recent progress in the field of DNA replication in early embryonic stem cells from the Torres-Padilla lab in Munich (e.g. PMID 35256805; PMID 38123678).*

Thank you for the suggestion. The same was also pointed out by the referee #1 and we have now included discussion on the replication mechanisms in 2CLCs investigated by Nakatani et al., from Torres-Padilla's group.

3. *p.6, lines 5-6, sentence is incomplete.*

Thank you for pointing this out. The sentence is now corrected (p17, line 2-3).

4. *p. 10, l. 10, typo in "endonegous"*

Thank you for pointing this out. This particular sentence is deleted to fit in the work limit.

5. *Some references are incomplete or in the wrong format, e.g. refs 30, 34, 40.*

Thank you for pointing this out. The relevant references are now presented properly.

6. *Typo in supplementary Table title.*

Thank you for pointing this out. The title is now reformatted.

Referee #3:

- 1) *The title of the article does not match the content. There is nothing in the paper about the high number of replication machinery, or any experiments attempting to quantify it.*

We agree that the title was rather inadequate for the content. We have now changed the title to "Pluripotent Stem Cells Maintain High Origin Activity and Slow Forks to Coordinate with Cell Cycle"

- 2) *One question that remains unanswered is what mechanism limits fork speed in pluripotent cells,*

or what mechanism speeds up the forks in the non-pluripotent cells, that is absent in the pluripotent cells. Do they have lower nucleoside concentrations due to downregulation of one or more metabolic enzymes? It would be very interesting to compare nucleoside concentrations between the types of cells used in this study and whether they change during S-phase progression. Or maybe the authors have a different hypothesis on the mechanism of the fork speed regulation. If it is impossible within the timeline of revisions, this should be discussed with the information available from the literature, and a brief schematic of the proposed model could be helpful.

Cross-comments from referee 2:

Regarding reviewer 3, point 2: I agree that comparing the nucleoside concentration of pluripotent vs mESCs would be interesting but it also may be a difficult task, as few laboratories have the biochemical tools to do it reliably. I agree that a better discussion of the possible reasons underlying the different fork speed would be useful

Thank you for these constructive comments and suggestions. Regarding nucleoside/nucleotide quantification, we have also been desperately seeking for ways to accurately measure them in different cell types and different cell cycle phases. dNTP pool size is critical for normal DNA replication, and is tightly regulated through feedback mechanisms. Thus, we need a sensitive method to measure dNTP pool size. However, as referee 2 has kindly cross-commented, we have also found that the nucleoside (especially the deoxy-nucleosides and deoxy-nucleotides) measurements are not trivial. Specifically, nucleoside/nucleotide measurements are carried out by mass-spec using millions of cells, which is difficult if one tries to collect substages of S phase with cell sorting. We can obtain measurements in the bulk population, but one has to take into account that dNTPs are produced only in the S phase and different cell types present different percentage of cells in S phase. One also needs to consider cell volume per condition and in different cell types, because mass spec only gives total amounts of nucleosides/nucleotides in the sample and do not provide information on concentrations per volume. We are currently seeking ways to understand the biology behind this using various approaches. Therefore, while we also think it is very important to measure nucleosides/nucleotides in future, we decided to share our finding now, focusing on the origin density and fork-speed control in ES cells and how fork speeding in ES cells affect their genome replication.

Regarding the question of how nucleosides molecularly work to accelerate fork speeds in ES cells and what exactly limits fork acceleration still remains to be uncovered. However, our results clearly show that ES cells do not experience replication stress that leads to fork pausing. Rather, through the assay suggested by referee 2, the primary cause of fork slowing is exceptionally high levels of origin activity and caused by the failure of replication forks. The reason for slow forks in early S phase of non-pluripotent cell may be different, as SNLs show increased level of fork asymmetry specifically in early S phase. The remaining question is how nucleosides can force fork acceleration in ES cells, but this topic lies beyond the scope of this paper and warrants further investigation in future studies.

Another question is why fork dynamics are regulated differently in pluripotent vs non-pluripotent cells. In MEFs, it is possible that the slow forks in the early S phase arise as a result of limited dNTP pool size. Our data showing elevated fork asymmetry specifically in early S phase of MEFs which are suppressed by addition of nucleosides, are consistent with the hypothesis. Thus, nucleosides may accelerate fork speed in early S phase of MEFs in part by increasing dNTP pool size. In ES cells, fork asymmetry is NOT observed in early S phase, which leads us to think that dNTP pool size is sufficient from the start of S phase. Pasero's group has shown clearly in yeast that S phase initiate before accumulating enough amounts of dNTPs, which causes fork pausing and activation of MEC1/ATR pathway leading to activation of dNTP production. It is possible that a similar mechanism might be operating in MEFs, but this is not the case in ES cells. We have added these points to the revised manuscript and also inserted a diagram showing our model in Fig. EV4C.

Dear Dr. Tsubouchi,

Thank you for the submission of your revised manuscript. We have now received the enclosed reports from the referees that were asked to assess it. Referee 1 still has a few more minor suggestions that I would like you to incorporate before we can proceed with the official acceptance of your manuscript.

A few editorial requests will also need to be addressed:

- Please correct the name of the conflict of interest subheading to "Disclosure and Competing Interests Statement"
- The Table of Content in the Appendix file is missing page numbers and the figure legend, please add.
- Please upload the Source Data as one (zipped) folder per figure.
- Table 1 needs to be called as such throughout the manuscript text ("Table 1" instead of "Table"); also the title in the Table file needs to be updated to "Table 1" instead of "Table".
- Please note that the exact p values need to be provided in the legends of figures 1a, f; 3a-c, e; 4a-b, d, f-g; EV 1a-b, d, f-g; EV 2a, c; EV 3c; EV 4b.
- Please note that information related to "n" is missing in the legends of figures 4a, d; EV 1b, g; EV 4b. Please specify the number for "n" and whether it represents biological or technical repeats. This also applies to the next point.
- Although 'n' is provided, please describe the nature of entity for 'n' in the legends of figures 3a-c, e; 4b; EV 2a-c; EV 3b-c.

I would like to suggest a few minor changes to the title and abstract. Please let me know whether you agree with the following:

Embryonic Stem Cells Maintain High Origin Activity and Slow Forks to Coordinate replication with Cell Cycle Progression

Embryonic stem (ES) cells are pluripotent stem cells that can produce all cell types of an organism. ES cells proliferate rapidly and are thought to experience high levels of intrinsic replication stress. Here, by investigating replication fork dynamics in substages of S phase, we show that pluripotent stem cells maintain a slow fork speed and high active origin density throughout S phase, with little sign of fork-pausing. In contrast, the fork speed of non-pluripotent cells is slow at the beginning of S phase, accompanied by increased fork-pausing, but thereafter fork-pausing rates decline and fork speed rates accelerate in an ATR-dependent manner. Thus, replication fork dynamics within S phase are distinct between ES and non-ES cells. Nucleoside addition can accelerate fork speed and reduce origin density. However, this causes miscoordination between the completion of DNA replication and cell cycle progression, leading to genome instability. Our study indicates that fork-slowness in the pluripotent stem cells is an integral aspect of DNA replication.

Please also add to the abstract (and may be the title) whether your work is based on mouse or mammalian ESC, or other or several species.

Referee #2:

Kurashima et al have revised their manuscript on the comparative study of DNA replication dynamics in mouse embryonic stem (ES) cells vs differentiated mouse embryo fibroblasts (MEFs). In my assessment of their original paper, I had asked the authors to try to go deeper into some of its mechanistic aspects, and to this aim I had suggested several major points and a few smaller issues. The authors have addressed adequately most of my questions with new experiments and discussion points. The manuscript has improved notably and its relevance in the context of other recent advances on this field is more justified.

I was asked to comment on how the authors have addressed the points raised by Referee #1, who was mainly concerned about lack of novelty in regards to another recent article that was not cited in the manuscript. This article (Nakatani et al, 2024, Nature) is now cited and discussed (p. 3). Other technical issues raised by this reviewer, such as the number of fork structures counted in the DNA fiber experiments, have been addressed experimentally.

In my opinion the article is now suitable for publication but there are still some minor adjustments to be made, as some of the sentences added in the revision lack clarity or have grammatical issues, e.g.

-p8, lines 14-15: detection of gH2AX per se is not necessarily indicative of DNA damage. It is also a marker of replicative stress.

-p 10, lines 13-14: "replication origins are programmed to license more origins" does not make sense.

-p 12, lines 23-24; description of IOD analysis needs revision, e.g. "DNA fibers presenting continuous (or almost continuous, with small gaps) presenting CldU-IdU-CldU...."

Referee #3:

The authors addressed my concerns and I believe the manuscript can now be published.

All editorial and formatting issues were resolved by the authors.

Dr. Tomomi Tsubouchi
National Institute for Basic Biology
38 Nishigonaka, Myodaiji
Okazaki, Aichi 444-8585
Japan

Dear Dr. Tsubouchi,

I am very pleased to accept your manuscript for publication in the next available issue of EMBO reports. Thank you for your contribution to our journal.

Yours sincerely,
